# Evidence for structural control of mare volcanism in lunar compressional tectonic settings

Feng Zhang [1], Alberto Pizzi [2], Trishit Ruj [3], Goro Komatsu [2,4], An Yin[5], Yanan Dang [6], Yang Liu [1,7] & Yongliao Zou[1]

One of the long-standing enigmas for lunar tectonic-thermal evolution is the spatiotemporal association of contractional wrinkle ridges and basaltic volcanism in a compressional regime. Here, we show that most of the 30 investigated volcanic (eruptive) centers are linked to contractional wrinkle ridges developed above preexisting basin basement-involved ring/rim normal faults. Based on the tectonic patterns associated with the basin formation and mass loading and considering that during the subsequent compression the stress was not purely isotropic, we hypothesize that tectonic inversion produced not only thrust faults but also reactivated structures with strike-slip and even extensional components, thus providing a valid mechanism for magma transport through fault planes during ridge faulting and folding of basaltic layers. Our findings suggest that lunar syn-tectonic mare emplacement along reactivated inherited faults provides important records of basin-scale structure-involved volcanism, which is more complex than previously considered.

Understanding how magma can reach the lunar surface under contractional tectonics is not just a matter of one scientific debate, it is also important for geophysical studies of tectonic mechanisms and the thermal evolution history of the Moon, considering that there is a close affinity between lunar volcanism and impact basin tectonics. The origin of lunar wrinkle ridges (WRs) has been attributed to tectonic compression[1-5] that leads to upper crustal shortening postdating the major mare emplacement during the Nectarian-Late Imbrian period (Supplementary Note 1) of lunar history (~4.0−3.2 billion years ago)[6]. Stress models from the load on lunar lithosphere[7] suggest that the WR-formation timing becomes younger towards basin margins (relative to near mare centers) with the thickening of the elastic lithosphere over time (up to ~75 km thick[8]). Magmas at such depth are very difficult to produce sufficient stress to overcome the lithostatic pressure of overburden and reach the lunar surface[9], particularly in a compressional geological setting (with load-caused flexure and contraction of the upper lithosphere) that is always considered unfavorable or even impossible for volcanism[10]. However, lunar scientists have long been puzzled by the clear spatial correlation between mare volcanism and WRs[11-14], which creates a dilemma as the latter is not normally associated with basaltic volcanism on Earth[15,16].

On Earth, volcanism has been typically associated with extensional regimes, though recent studies highlight how it can also develop in compressional tectonic settings, comprising both strike-slip and contractional deformation[16-24]. Concerning strike-slip faulting, this is considered to be a rare phenomenon on one-plate solar-system bodies such as the Moon because of the lack of plate boundaries[8,25]. Nevertheless, strike-slip faults have been theoretically predicted to be an important element of tectonic patterns associated with contractional WRs[3,25,26], the pattern and trend of which contain information of

[1]State Key Laboratory of Space Weather, National Space Science Center, Chinese Academy of Sciences, Beijing, China. [2]Department of Engineering and Geology, Università d'Annunzio, Chieti-Pescara, Italy. [3]Institute of Space and Astronautical Science (ISAS), Japan Aerospace Exploration Agency (JAXA), 3–1-1 Yoshinodai, Sagamihara, Kanagawa 252–5210, Japan. [4]International Research School of Planetary Sciences, Università d'Annunzio, Pescara, Italy. [5]Department of Earth, Planetary, and Space Sciences, University of California, Los Angeles, CA 90095-1567, USA. [6]National Key Laboratory of Microwave Imaging Technology, Aerospace Information Research Institute, Chinese Academy of Sciences, Bejing, China. [7]Center for Excellence in Comparative Planetology, Chinese Academy of Sciences, Hefei 200083, China. ✉ e-mail: zhangfeng@nssc.ac.cn; alberto.pizzi@unich.it

underlying preexisting topography (e.g., buried impact structures and the subsurface architecture of faults and fractures)[2,3,26–29]. Observational analysis and mathematical models of WRs support their formation by shallow-depth thrust faults[3] (<~5 km, see Summary in Watters (7)). Thus, the WRs formation associated with shallow thrust and/or strike slip-like faulting are difficult to reconcile with mare flooding events because their source regions are deep-rooted in the lunar mantle[9,30].

As a long-standing unresolved issue, it is crucial to correctly evaluate the relationship between the tectonic structures that could be associated with the WR-formation (i.e., compatible in terms of stress, geometry, and chronology) and their potential to drive volcanic eruptions. In this work, we conducted a photogeologic mapping of 30 mare volcanism-tectonic interaction zones across the globe and then focused here on a well-preserved type area in detail where mare volcanism occurred in a compressional tectonic setting hosting WRs. We present a model considering a combination of WR-formation and mass-loading tectonics to reconcile coeval WRs formation and fault-facilitated lunar mare volcanism, which is of key importance for the understanding of lunar thermal evolution, mascon loading tectonics[8,31], and the ancient igneous activity of the Moon[32,33].

## Results

### Volcanic eruptions at the transfer zone linking contractional structures

Mare Imbrium (37°N, 18.5°W), a mascon mare basin (Fig. 1a, b), is the second-largest (after 2400-km-diameter South Pole-Aitken), multi-ring basin on the Moon. It has three prominent rings determined by measurements from topography and the Bouguer gravity anomaly[34] (dashed circles, Fig. 1). The Lambert-SE structure is located just southeast of the Lambert crater in southern Mare Imbrium (solid circle, Fig. 1). The intermediate ring of the basin cuts across the structure (Fig. 2a). Geologically speaking, a series of normal faults (also known as ring faults) resulting from the inward collapse of the basin interior are commonly associated with these ring structures[8,35,36]. The faults may terminate within the lower crust, transitioning into a basal décollement and/or continue by cutting the entire crust[37], even some extending into the upper mantle[38] as revealed by locating shallow-focus moonquakes (~50–200 km in depth[39]) distributed around the nearside basins, which are thought as resulting from lunar fault activity[5,40].

Lambert-SE has a quasi-circular form constrained from the positive gravity anomaly (Fig. 2b and Supplementary Fig. 1), and it is interpreted to be a potential ~120-km-diameter buried impact crater[41]. At its central part, a pit-like feature (green triangle, Fig. 2) is observed and is interpreted to be of volcanic origin[14,42] due to its irregular shape, very steep-slope (nearly vertical) wall, broad flat floor, and very low depth/Diameter value (d/D: 150/3200 m; ~0.05 vs. ~0.2 for lunar simple impact craters ranging between 400 m and 10 km in diameter[43]) (Supplementary Fig. 2a). This purported volcanic crater is located on the top of a WNW-striking ridge which connects two major sets of NNW-striking *en echelon* WRs, here interpreted as a transfer zone (dashed black ellipse, Fig. 2a), an oblique zone between two parallel adjacent contractional WRs able to transfer the deformation through oblique/strike slip-like motion[3]. The fan-shaped deposits and flow front scarps (Fig. 3) indicate volcanic flows emanating from the crater vent[14]. To its southeast, a sinuous rille originates and runs towards the southwest and south with a total length of ~86 km (R1, Fig. 2).

### Structural control of mare volcanism in compressional tectonic settings

At the Lambert-SE region, the maximum horizontal compressional stress axis is concentric to the basin center (or parallel to the basin rings, i.e., near E-W), as can be inferred from the spatial orientational pattern of the major sets of thrust faults and folds at both sides of the transfer zone (i.e., ranging from N-S to NNW-SSE, Fig. 2 and

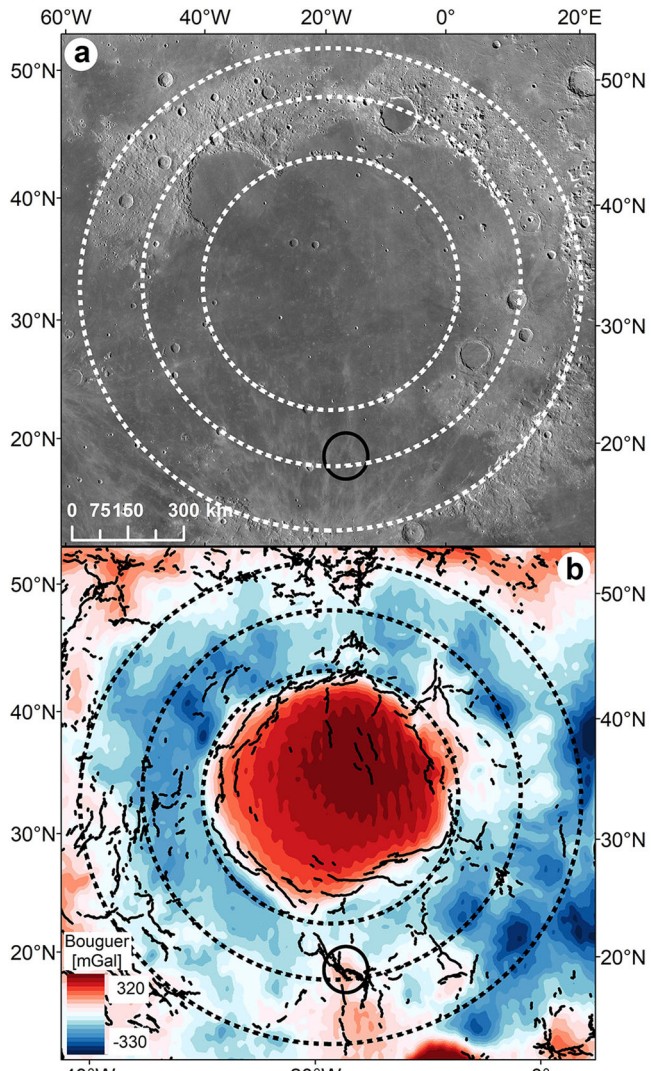

**Fig. 1 | Geological background of the Imbrium basin. a** LROC WAC mosaic for the multi-ring Imbrium basin. The black solid circle overlapping the second basin ring in the south illustrates the location of the Lambert-SE Structure. **b** GRAIL Bouguer gravity anomaly map for the Imbrium basin. The three complete Imbrium rings (black dashed lines) are determined from topography measurements and the GRAIL Bouguer gravity anomaly[34]. The black solid lines indicate the major wrinkle ridges. LROC lunar reconnaissance orbiter camera, WAC wide-angle camera, GRAIL gravity recovery and interior laboratory.

Supplementary Fig. 3). The nearly E-W compression, indeed, is coherent with the linear pattern of the N-S oriented WRs and with the left-stepping arrangement of the NNW-striking, and, particularly, NW-striking WRs. This same stress field would lead to reactivation as oblique/strike-slip faults of preexisting discontinuities (ring faults) that formed at an acute angle to the orientation of the principal horizontal stress axis (Supplementary Fig. 3). Now, as these reactivated ring faults are inherited deep-seated and high-angle faults, they might have served as conduits for mantle magma ascent and hence eruption at the Lambert-SE transfer zone (Fig. 3). The occurrence of releasing step-overs along the sinistral transfer zone provided zones of localized extension which may have further facilitated the migration of magma towards the surface, as supported by the presence of the interpreted volcanic crater that occupies just one of these areas (Supplementary Fig. 4). The surface source region of the sinuous rille R1 is also well situated in the transfer zone (Fig. 2), implying the volcano-tectonic evolution of the area. Furthermore, the elongated shape of the

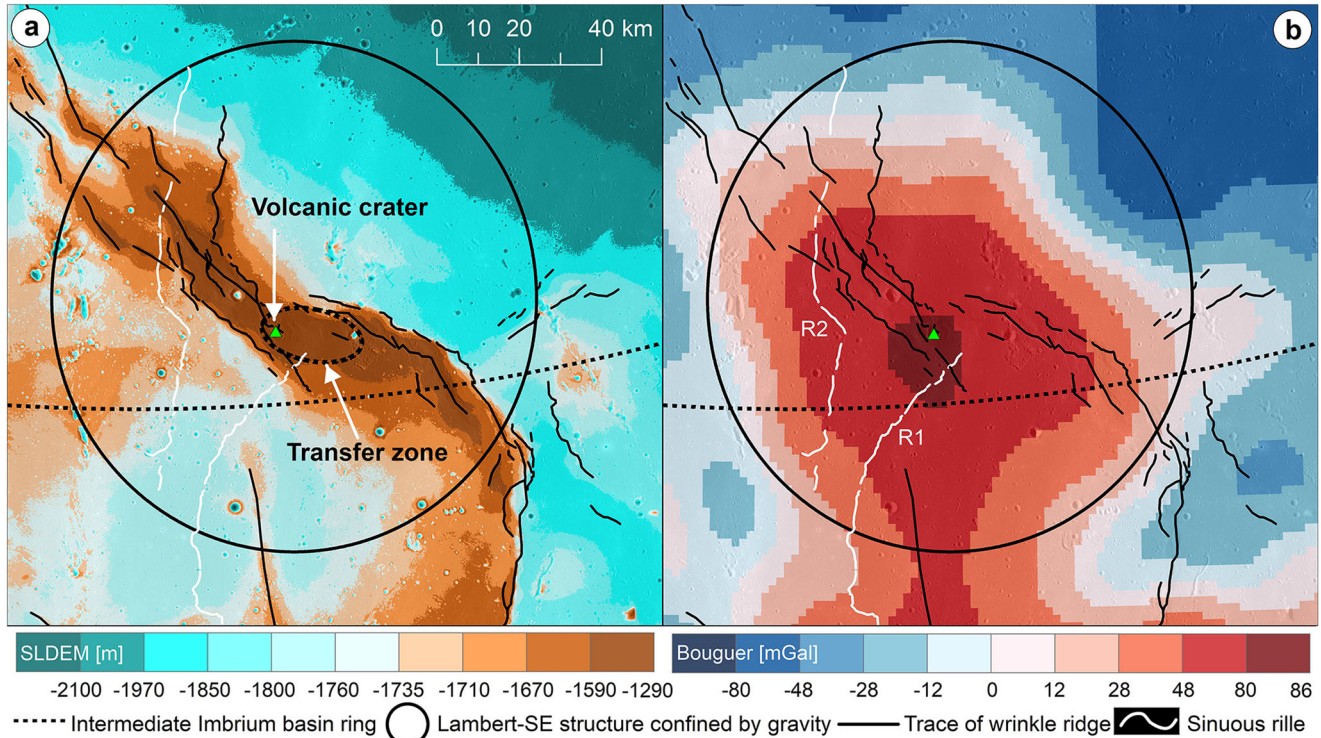

**Fig. 2 | Topographic and gravitational characteristics of the Lambert-SE region. a** SLDEM2015 (SELENE and LRO DEM 2015) topographic map for the volcanic crater (the green triangle) and the head of the sinuous rille R1 located in the transfer zone linked with two wrinkle ridges displaying a left-stepping pattern. **b** The volcanic crater is also situated in the center of the positive Bouguer gravity anomaly, which is used to define the outline of the Lambert-SE structure[41]. LRO lunar reconnaissance orbiter, DEM digital elevation model.

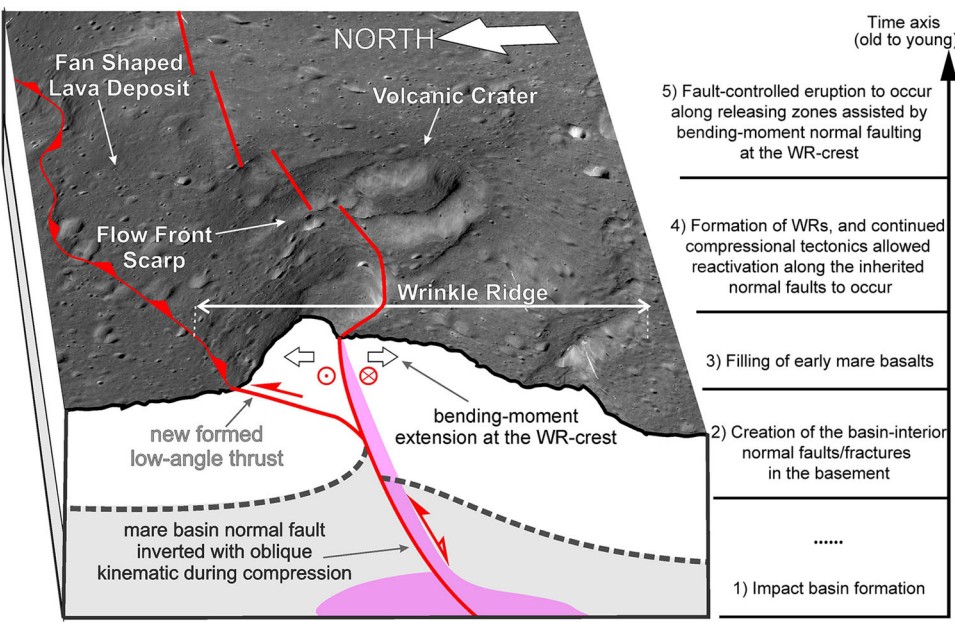

**Fig. 3 | Sequence of events for structural control of mare eruption.** Kaguya TC map draped on TC-derived DEMs (-10 m/pixel, 6× vertical exaggeration) for the detail of the volcanic crater (21.30°N, −17.39°E; -3.2 km in diameter for scale) and its geologic background with contractional wrinkle ridges. The schematic (not to scale) vertical section in the foreground shows the reactivation during compression of the deep-rooted preexisting normal fault with oblique-slip kinematics, as expected at the transfer zones. At the shallow crustal level, deformation is partitioned between newly formed low-angle thrusts (and folding) and oblique/ strike-slip faulting, commonly along the preexisting discontinuities (i.e., strain partitioning). The overall high-angle oblique/strike-slip and inverted normal fault structures might have served as conduits for mantle magma ascent and eruption at the Lambert-SE transfer zone. The flowchart at the right indicates a general timeline of the major events described in this model. TC Terrain camera, DEM digital elevation model.

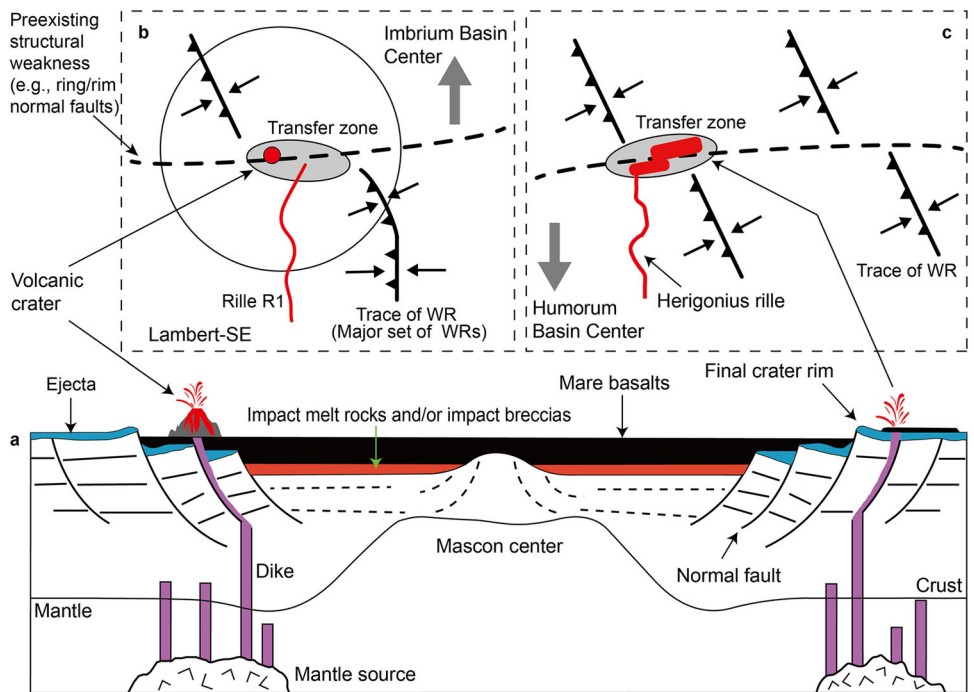

**Fig. 4 | Sketch maps for the syn-tectonic volcanism in the Lambert-SE and Herigonius rille areas. a** Normal faults formed along the impact basin rings/rims provide an extensional tectonic setting favorable for magma ascent. The Lambert-SE structure superposes directly on these normal faults adjacent to the intermediate ring of the Imbrium basin. **b** The compressional stress field with a maximum horizontal axis oriented approximately at right angles to the major wrinkle ridges (black arrows) leads to horizontal movements along the oblique transfer zone, linked with them, whose localization is controlled by the underlying basin ring normal faulting zone (i.e., preexisting structural weakness), hence further favoring the localized mare volcanism. **c** The strike slip-like fault control of mare eruptions for the Herigonius rille formation in southern Procellarum (see Fig. 5 for location). The direct observational evidence is inferred from the high-resolution imagery and topographic data (Supplementary Fig. 6).

depression within which the volcanic crater (Fig. 3) is bound by the fault geometry (Supplementary Fig. 2b, c), suggests a structural fault-controlled flow of magma. Therefore, we hypothesize that the bending moment within the outer arc of the WRs provided a further source of extension capable of producing open fractures and normal faults along the crestal zone of the fold[44,45], thus facilitating the ascent of magma to the surface.

The gravitational measurements reveal a thinned lunar crust less than 20 km for Lambert-SE (Supplementary Fig. 5; ref. [46]), and this supports the hypothesis of the presence of a zone of crustal weakness favorable for upward movement of pressured magmas (e.g., dike propagating). A requirement of mare volcanism here is that a magma reservoir in the upper mantle should have already existed or begun to form before the initiation of local ridge-formation tectonics. Mare basaltic rocks have liquid densities[47,48] of between ~2750 and 3100 kg m$^{-3}$, which have a positive buoyancy within the upper mantle (~3200–3400 kg m$^{-3}$; ref. [49]) and are even slightly lower than the lower crust, which is composed predominantly of norite[47] with a density of ~3100 kg m$^{-3}$, favoring ascent of the basaltic liquids through the upper mantle and/or lower crust. The buoyantly rising magma may have been trapped at the crust-mantle boundary or the interface of a level of neutral buoyancy[9,50]. Thus, hot melt stored at depth preferentially flows toward the thinned crustal region with extensional features and/or high-angle oblique/strike-slip faults, causing surface eruptions.

**A tectonic scenario for mare volcanism**

The transfer zone in the Lambert-SE area is located in the overlap region where the surface WRs intersect underlying basin-concentric ring/rim normal faults[51,52], along which post-impact tectonics-induced movements would occur. Thus, a sequence of events accounting for the apparently contractional tectonic-controlled volcanism is predicted: (1) Basin-concentric normal faults that reside in the lunar crust formed during the period of post-impact tectonic adjustment (Fig. 4a); (2) Early mare basin filling events resulted in a decreasing trend of the thickness of mare basalts from the mascon center towards the basin rim[53,54]. The central mare load may have exerted tension on the lithosphere, leading to dilation of the basin-concentric faults, which provide ideal geological settings favorable for magma ascent[55], resembling narrow buried lava-flooded rift valleys in the underlying feldspathic/noritic crust[56]; (3) In chronological order, after the emplacement of early mare basalts, the adjustment of mascon and basalt loading enables the Lambert-SE region shifting into a geologic setting dominated by compressional tectonics. In this new tectonic scenario, possibly the preexisting buried normal faults may have exerted a structural control over the location and geometry of WRs, as inferred from the regional arrangement of WRs characterized by radial and concentric patterns[56]; (4) At the intersection place between the major WRs and underlying concentric normal faults of weakness, post-mare tectonics allowed reactivation along the inherited normal faults to occur (Fig. 4b), modifying the strike of WRs at shallow crustal level (also see the Herigonius-rille-formation eruptions; Fig. 4c and Supplementary Fig. 6). The transfer zones in the Lambert-SE and Herigonius-rille areas may provide evidence to support the movement-induced oblique slip versus opening along fault segments for later eruptions.

Lunar WRs often display regularly or irregularly spaced faults and ridge segments of different orientations and opposing asymmetry[3]. Under some circumstances, these complex WRs geometric patterns may also reflect a pre-compressional anisotropy of the crust due to preexisting structures that have been reactivated during compression. The orientations of reactivated older structures, indeed, are unlikely to be ideal for either pure strike-slip or pure dip-slip motions; thus, oblique-slip displacements on reactivated faults are typically important within transfer zones and fault bends[27,57,58]. Therefore, alongside

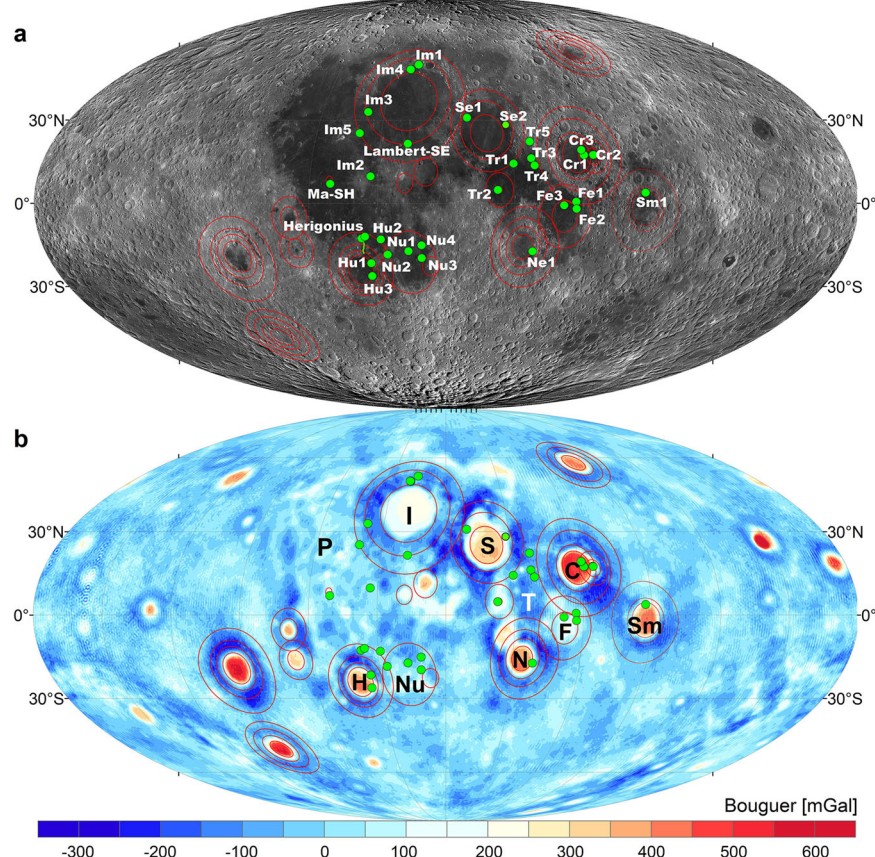

**Fig. 5 | Global distribution of the 30 compressional tectonic-volcanism cases.** The locations (green points) where interactions of mare volcanism and contractional wrinkle ridge-formation tectonics occurred are shown in **a** LROC WAC mosaic and **b** GRAIL Bouguer gravity map. Red circles mark the basin rings defined by a combination of topography and gravity[34]. Aside from the Lambert-SE, the geologic details of other 29 locations with similar cases of volcanism within a dominantly contractional tectonic regime are shown in Supplementary Figs. 6–34. The Mollweide projection is used here. I Imbrium, S Serenitatis, C Crisium, T Tranquillitatis, F Fecunditatis, N Nectaris, P Oceanus Procellarum, H Humorum, Nu Nubium, and Sm Smythii, LROC lunar reconnaissance orbiter camera, WAC wide-angle camera, GRAIL gravity recovery and interior laboratory.

the folds and thrusts, oblique/strike-slip and extensional structures can coexist, accommodating the non-coaxial component of the deformation. Such structural association is noticed in and around the Lambert-SE transfer zone, where oblique-slip displacements on the deep-rooted reactivated normal faults are accommodated at a shallow crustal level by both WRs and low-angle thrusting as well as high-angle strike-slip faulting (strain partitioning). This scenario can explain through a simple compressional deformation the origin of the relief at the transfer zone produced by thrusting and folding of the upper mare units and contemporaneous oblique/strike-slip motion along the inherited mare basin normal faults, which propagate up to the surface at the crestal zones of the growing WRs. The connection of the oblique/strike-slip fault with the underlying preexisting normal fault plane hence is likely to provide an ideal high-angle conduit for later eruptions.

## Global distribution

Aside from Lambert-SE, other 29 locations where lunar mare volcanism and contractional WRs tectonics are related have been identified (Fig. 5 and Supplementary Figs. 6–34). These cases provide strong evidence to support the subsurface structural control of mare volcanism in lunar compressional tectonic settings. Most of them have a spatial correlation with basin ring/rim normal faulting zones well inside the influence of mass-loading tectonics. Though the crustal normal faults or fractures were flooded and buried by volcanic materials, some were later reused and served as structural weaknesses for mare eruptions, which

postdated or occurred with WR-formation compression tectonics concurrently. For example, the sinuous Herigonius rille in southern Procellarum is located within the influence of the mass-loading tectonics of Mare Humorum (Fig. 5). Its source head shows two elongated, *en* echelon-patterned depressions (Fig. 4c and Supplementary Fig. 6) that are interpreted to coincide with the preexisting structural weakness in the normal faulting basin ring/rim zone, possibly reactivated later with lateral slips.

The 30 compressional tectonic-volcanism cases can be divided into four classes based on their eruption sites and WRs concentric or radial to the basin center. The first class includes the Lambert-SE and Herigonius rille regions (Fig. 4). The eruption centers are located at the intersection areas between WRs and inferred basin ring/rim structure zones hosting concentric normal faults or fractures in the basement. We here defined the intersection areas as transfer zones where volcanic eruption centers are located. Later reactivation of subsurface normal faults with lateral slips (horizontal movement of rocky bodies on both sides of normal faults) by post-mare compression provides conduits for magma transport.

The second class points to the cases in Supplementary Figs. 7–12. Similar to Lambert-SE and Herigonius, their eruption centers are located at the intersection points between WRs and sub-mare basin ring/rim normal faults or fractures. However, no surface clue indicating horizontal movement at the eruption sites is observed. A direct link between sub-mare normal faults and intra-mare thrust-derived faults/fractures may have provided pathways for later eruptions. Some cases

(Supplementary Figs. 13–22) can be divided into the third class, with their eruption centers situated on the broad arch of WRs. These WRs are located at the basin/crater ring/rim zones where normal faults and extensional fractures preferentially occur. Nearly all of them are concentric to their host basins or buried impact structures. For the abovementioned three classes, we interpret them as the compressional tectonic-influenced mare volcanism that has been fed by basement-involved normal faults/fractures at the basin/crater ring/rim zones, which provide pathways for the magma transport and dike propagation to reach the surface and hence erupt.

The others (Supplementary Figs. 23–34) are classified into the last group. These cases display a spatial relationship between volcanic eruption centers and WRs. Their volcanic edifices are bounded by or situated on or near WRs. Unlike the classes 1 to 3 mentioned above, these WRs do not show such a clear relationship with large-scale basin/crater ring/rim structures. Nevertheless, their formation and development might have been significantly influenced by localized underlying preexisting topography. Currently, whether these volcanic eruptions formed before or after the WRs cannot be confidently confirmed. We keep this issue to be further investigated and solved through future efforts.

## Discussion

On the Moon, reactivation of preexisting faults in compressional tectonic domains results in wrinkle ridges, and subsequent structural adjustment triggers volcanic eruptions in mare basins. Here we provide evidence that the development of the lunar WRs investigated in this study has been strongly controlled by the reactivation of preexisting mare basin faults (mostly extensional, Fig. 6a) during later compression (positive inversion). First, WRs are associated with mare basalts since they commonly occur in both mascon (mass concentration with positive gravity anomaly[59]) and non-mascon large-scale mare basins at their interiors, near-margins, and/or

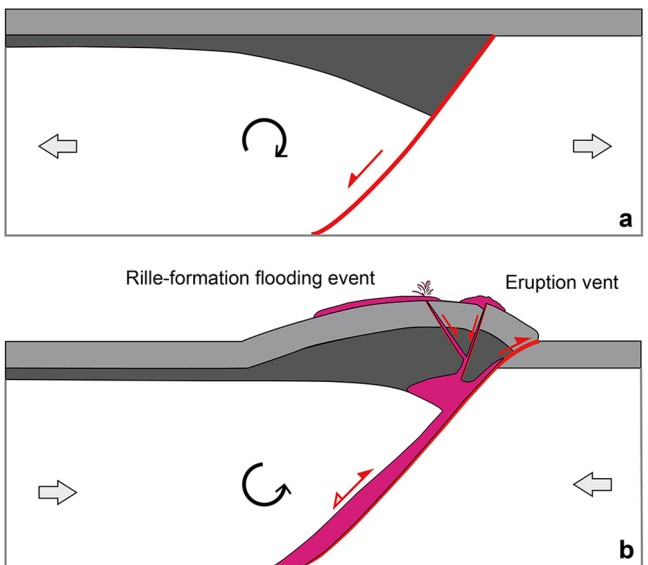

**Fig. 6 | Tectonic inversion origin of lunar wrinkle ridges.** A simplified schematic model showing the two stages of **a** extension and **b** inversion with associated volcanism and WR-formation. Inversion causes the reactivation of the mare basin normal fault that propagates upward into the previously unfaulted post-mare unit (light gray color) as a new thrust fault, and folding of the syn-mare unit (dark gray color). The magma acts as a lubricant along the faults, reducing fault strength, hence facilitating major slip and fully fault reactivation under a dominant compressional stress field. The magma (pink color), favoring the full reversal of the deep-rooted normal fault, accumulates in the core of the WR fold[66], where bending moment fractures and normal faults can make possible the ascent of magma to the surface. Not to scale. WR wrinkle ridge.

outside of them[1,60,61], as well as within and bordering anomalies interpreted to be mare basalt-filled rift valleys[56]. Second, their preferred concentric and, subordinately, radial orientations with respect to the basins show a strong correlation with the preexisting structural pattern of faults expected from the former post-impact (mascon) tectonics[8,31–33]. Third, the dimensions and shapes of WRs are very similar to those characterizing several folds within extensional basins which experienced tectonic inversion on Earth. Lunar WRs generally have an elevation of up to a few hundred meters, a length of tens to hundreds of kilometers, and a width of tens of km. The typical profile sections represent a broad arch often superposed by narrow ridge/s[1,2,28,29]. In particular, the arch is a broad topographic rise, commonly asymmetric in cross-section, having an extremely long limb with a very gentle slope, while the other limb is steeper and very short. These dimensional and geometrical features, indeed, are typical of many folds occurring in the Southern North Sea (northern Europe), one of the most studied examples of tectonic inversion of a previous extensional basin[62].

The tectonic inversion of inherited normal faults oriented obliquely with respect to the direction of the maximum horizontal stress axis ($\sigma 1$), as in the case of the Lambert-SE transfer zone, maybe a rather widespread phenomenon on the Moon (in particular in zones of the mascon basin rings/rims with distributed normal faults), as it is also supported by other examples shown in the Supplementary information (e.g., Supplementary Figs. 6–22). Analog modeling and field examples on Earth, indeed, show that oblique compression favors oblique-slip reactivation of normal faults that are unlikely to be reactivated in an orthogonal compression due to the characteristic steep angle of normal faults[62–64], i.e., ~60°.

However, based on the strong evidence that contractional WRs largely reflect the articulated pre-shortening setting of faulting, both internally and at the edges of mare basins and valley rifts, it is reasonable that inversion tectonics should be a more generalized deformation mechanism in the contractional history of the Moon and it should not be relegated just to cases of transfer zones. Then, assessment of which mechanism makes high-angle inherited structures orienting orthogonally to the shortening direction (e.g., basin ring/rim normal faults) to be fully reactivated considering also the low budget of compressional stress conditions[65], remains challenging.

Sandbox modeling studies on magmatism during inversion tectonics have shown that the fluid acts as a lubricant along the faults, reducing fault strength, hence facilitating major slip and fully fault reactivation under a dominant contractional stress field[66]. Moreover, the same authors pointed out how magma migration in inverted basins is typically controlled by the distribution of inherited structures which create the main emplacement pathways during their reactivation, and how it promotes the development of anticline structures, similar to the lunar WRs, whose core magma accumulates near the surface (Fig. 6b).

In such a geological setting, it is possible that overpressured magma may leak onto the lunar surface when local situations of extensional stress arise, produced by the bending moment along the crestal zone of the WR (Fig. 6b), as shown in the example of Lambert-SE (Fig. 3). In this context we believe it is possible to explain many of the other investigated cases of volcanism related to WRs as illustrated in the Supplementary information (Supplementary Figs. 6–34) and above all to underline how tectonic inversion possibly represents a common lunar contractional deformation mechanism than hitherto understood. We believe that additional examples of volcano-tectonic interactions resembling the reported 30 cases will be identified if future global searching efforts are performed. Furthermore, a larger population of mare wrinkle ridges of tectonic inversion origin is expected to have occurred because most of them may have not been accompanied by mare volcanism.

In summary, we interpret the development of transfer zones, tear faults, and, more generally, sharp changes in the axial trend of WRs, as commonly related to the structural control exerted by preexisting basement heterogeneities and weaknesses. In these zones of non-coaxial contractional deformation, reactivation of older faults involves an important component of oblique-slip kinematics, which at shallow crustal level, seems to be accommodated by partitioning of the displacement components, through (i) strike-slip faulting along, possibly reactivated[67], high-angle fault planes, and (ii) newly formed low-angle thrusting. Therefore, in this study, the identified 30 cases with eruption centers linked to compressional tectonics would provide useful information concerning their host basins' infilling and cooling history, enabling to draw a clear picture of the thermal evolution history of the Moon after its ancient impact basins had formed[68,69].

The WRs in the Lambert-SE region are situated well within the influence of the Imbrium mascon and correlate with deep-seated intrusions detected by the GRAIL gravity data[60]. The basaltic units in which the Lambert-SE lies were dated to be ~3.5 Ga by ref. [70]. According to the cross-cutting relationship, the WRs within Lambert-SE should have formed after the emplacement of the host basalts. This means that oblique/strike-slip faulting in the transfer zone and resulting volcanic eruptions at ~3.5 Ga or later (~0.42 Ga after the basin formation of ~3.92 Ga[71]) are consistent with the global survey of lunar WRs with average ages between 3.5 and 3.1 Ga[6]. The WRs (aged ~3.1 Ga on average[6]) interior of the Imbrium basin, typically occur radial or concentric to the basin center, suggesting their formation possibly resulted from subsidence and flexure due to iso-static compensation[8,31,33,72] subsequent to the major emplacement of mare basalts ~3.3 Ga ago[70].

The Lambert-SE case would be an excellent and illustrative example of contractional deformation of mare basalts acting upon underlying crustal normal faults (preexisting structural weaknesses) formed during the post-impact tectonic adjustment. The upper ~20 km of the lunar crust may have been significantly fractured by extensive impact cratering[73], which facilitates magma rise. This is consistent with the predicted thermal tectonics on one-plate planets where much of the thermal stress is due to differential expansion and/or contraction of the interior relative to the lithosphere and large-scale faulting of the lithosphere is the expected failure mode[32]. The pressured basaltic magma in the lunar upper mantle can rise to the surface via these normal faults and their overlying mare weaknesses created during the basin formation and subsequent self-adjustment tectonics. However, the volcanic eruptions within the Lambert-SE domain overlap the periods of wrinkle ridge growth and development, suggesting that igneous activity and compressional tectonics could have occurred simultaneously, rather than in a strict chronological order. The interpretations of these features, therefore, provide important implications for the understanding of mare volcanism, lunar thermal evolution, and Moon-interior geodynamic processes.

In this work, volcanic eruption centers associated with contractional wrinkle ridges in the lunar maria are viewed in the context of their particular tectonic environments. On the one hand, underlying preexisting topography (in particular, buried impact structures and the architecture of faults and fractures) has an important influence on the growth and development of wrinkle ridges during later sequences of mare filling and cooling events. On the other hand, regional mass loading-caused compressional tectonic activities exert structural control of mare eruptions by opening intra-mare channels connected to sub-mare normal faults. This implies the formation of wrinkle ridges concurrent with mare volcanism; thus volcanism in zones of active compression on the Moon may be more frequent than previously thought.

As we continue to explore and make new geologic and geophysical observations of volcanic features and structural elements of mare wrinkle ridges, we develop new insights into the magmatic and tectonic processes that led to the emplacement of basaltic lava flows and accommodation of oblique/strike-slip-like kinematics associated with wrinkle ridges. During the Moon's early stages of large impact cratering, lithospheric thinning, and extension are accommodated in the crust by excavation and normal faulting. Later global or regional scale compressional stresses dominate basin-interior thrust faulting and contractional deformation, producing and making wrinkle ridges one of the most noticeable traits of global volcanic plains. Local oblique/extensional deformation within the lithosphere can be related to strain partitioning during compressional tectonics due to the control and reactivation of a complex pattern of preexisting buried extensional structures. Under such conditions, high-angle oblique/strike-slip faulting and associated extension along releasing zones assisted by bending-moment normal faulting at the WR-crest coeval and stress-compatible with the contractional wrinkle ridge tectonics appear to provide an efficient pathway for ascending magma. Thus, volcanic activity overlapped the periods of wrinkle ridge growth and development.

The formation of contractional wrinkle ridges concurrent with mare volcanism reveals that gravity-caused tectonic adjustment due to mass loading may have been the dominant factor to control the evolution of the basin tectonics and late-stage mare fillings, though the thermal stress due to differential expansion and contraction of the interior played an important role[32]. This may be also the reason why nearly all wrinkle ridges are confined to mare basalts[60] (i.e., the basin interior), only with very few extending into the nearby highland regions[3,13,29]. Conventionally, the wrinkle ridge style of deformation is confined solely to the mare regions because the mare basalts are multilayer sequences, therefore, deform as a multilayer rather than massive material such as the highlands[5]. This is clearly demonstrated by the wrinkle ridge-lobate scarp transitions that occur at mare-highland contacts[5]. Basin-localized extension ceased ~3.6 Ga ago[74]. The overlapped periods of the cooling-dominated WR compression tectonics and continued mare volcanism between ~3.0 and 3.6 Ga may support and be well consistent with our proposed model and conclusion: during this transition period, the occurrence of late-stage mare volcanism (some but not all) is concurrent with mare wrinkle ridge-formation tectonics, which provides vertical channels for magma ascent. Thus, the simple order of the resulting compressional stress to form wrinkle ridges following the basin filled with approximately several kilometers of basalt needs to be reassessed. The apparent relationship that all the mare wrinkle ridges should postdate their host mare units may be not always true because wrinkle ridge-formation compressional tectonics can provide mare weaknesses (e.g., thrust faults and associated extension at the WR-crest) for concurrent or even later mare volcanism, which is fed by magma transport through the basin basement-involved crustal normal faults/fractures.

## Methods

The geologic background of the study areas is provided by photogeologic mapping and interpretations of features and units observed in orbital imagery of the lunar surface, topographic maps and profiles, geophysical data, and existing geological maps (Supplementary Note 2). Wrinkle ridges and volcanic features are mapped using lunar mission data collected by the Lunar Reconnaissance Orbiter (LRO; ref. [75]) and SELENE (Kaguya)[76] missions (Supplementary Note 3). The age relations among the mapped units are revealed through their superposition and cross-cutting relations[13]. Wrinkle ridges are digitized with ArcGIS 10.8 using the method described by ref. [77] (Supplementary Note 4). In addition, the Gravity Recovery and Interior Laboratory (GRAIL; ref. [78]) gravity models of lunar crust[46] (Supplementary Note 5) are used to investigate subsurface crustal conditions. The description of maps and DEMs used for the regional geologic details of the 29 locations as shown in Supplementary Figs. 6–34 is also

given (Supplementary Note 6). Please refer to the text in the Supporting Information for more details on an extended methodology of geologic mapping and the data sets available.

## Data availability
Data used in this work are archived in the Geophysics Node of the Planetary Data System (PDS, http://pds-geosciences.wustl.edu/dataserv/moon.html) and the JAXA lunar orbiter KAGUYA/SELENE (http://darts.isas.jaxa.jp/planet/pdap/selene/). The URLs for the data acquired from different missions and instruments are also provided in the Supplementary file.

## Code availability
The GRAIL gravity data model GL0420A used to derive the lunar crustal thickness can be accessed at https://doi.org/10.5281/zenodo.997347[46].

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

## Acknowledgements

F.Z. was funded by the National Key Research and Development Program of China (2022YFF0503100), the National Natural Science Foundation of China (12273044), the Key Research Program of the Chinese Academy of Sciences (KGFZD-145-2023-15), and the Science and Technology Development Fund, Macau SAR (0049/2020/A1). T.R. acknowledges the support of JSPS KAKENHI No JP17H06459 (Aqua Planetology).

## Author contributions

F.Z. conceptualized the initial study and developed the methodology for mapping and investigation. F.Z., A.P., and T.R. were responsible for conducting the data analysis, developing the model, and leading the writing and visualization of the results. F.Z. led the data analysis for the observations with assistance from A.P., T.R., G.K., A.Y., Y.D., Y.L., and Y.Z. All the authors contributed to the writing, interpretation, and analysis.

## Competing interests

The authors declare no competing interests.
