## [Peer Review File · Nature Communications]

Evidence for structural control of mare volcanism in lunar compressional tectonic settingsEditorial Note: Parts of this Peer Review File have been redacted as indicated to remove third-party material where no permission to publish could be obtained, and to maintain anonymity.

REVIEWER COMMENTS

Reviewer #1 (Remarks to the Author):

This manuscript investigates associations and argues for a link between tectonism and volcanism in a portion of Mare Imbrium and generalizes more broadly to other locations across the lunar maria. Specifically, compressional tectonism represented by wrinkle ridges is proposed to have occurred both contemporaneously with and after volcanism, and that the tectonic structures provided conduits at which magma might ascend. The manuscript also hypothesizes that oblique and/or lateral motion on some faults occurred under the control of pre-existing weaknesses in the lunar crust.

The paper is generally well written, and is an interesting case study with new examples of vents + wrinkle ridges that I had not specifically seen before. However, I do not believe that the broader concepts of association between volcanism and tectonism are particularly new, and any evidence for strike-slip faulting is lacking.

Wrinkle ridges that ponded lava flows and also continued to deform subsequently have previously been documented back to the post-Apollo era (e.g., Bryan, 1973; Schaber, 1973; Schaber et al., 1976). Similar relationships have also been observed with terrestrial tectonism/volcanism in the Columbia River basalts (Reidel, 1984; Watters, 1988).

I find intriguing the hypothesis of a "transfer zone" between adjacent segments of wrinkle ridges as being strike-slip faulting, but no observational evidence supporting strike-slip faults is presented. Such evidence might include measured lateral displacements across surface features, diagnostic morphology assessment, or numerical modeling of resulting surface landform reliefs/displacements. No distinction is made between discrete faults vs a more distributed shear zone, which one might expect to have significant implications when suggesting these proposed faults as conduits for magma ascent.

Wrinkle ridges are complex features that may or may not necessarily include one or multiple narrow crests superimposed atop the structure, which can and often do represent antithetic faults. An example is shown in Figure S5.A, where the main wrinkle ridge crest to the southeast reaches the vent. Notably, the northwest trace as shown in S5.D has higher relief (its hanging wall) on the northeast side as seen in S5.B, indicating it is actually an antithetic fault with opposite vergence to the fault trace drawn on the southeast side of the vent, and with the opposite vergence to what it is shown as having in the figure – meaning it is not the same continuous fault surface expression offset laterally as indicated in S5.C. This example could be explained as the antithetic ridge crest forming to the northwest after and not propagating through the vent, without any strike-slip motion required. There is also a second more northerly trace not drawn in S5.D but where the zigzagged arrow is covering it up in S5.C, that appears would be continuous with the southeastern fault trace if the vent did not superpose it, further suggesting no lateral/strike-slip motion. Similarly, in Figure 2A, although the narrower superimposed ridges are largely absent in the transfer zone, the broader arched surface expression of the wrinkle ridge is still present throughout the transfer zone connecting the two segments, suggesting no large-scale changes in the deeper structure (other than it simply being a non-planar fault) that would require distinct strike-slip faulting.

Another consideration not mentioned in this paper is that although wrinkle ridges occur in the lunar mare and other planetary basaltic plains, this surface morphologic classification does not necessarily require its underlying faults be controlled by or have a driving force from volcanism itself. Rather, wrinkle ridges are likely a form of surface expression resulting from compressional faulting and folding (possibly but not necessarily driven by volcanic emplacement) in a layered material consistent with what layered flood volcanism presents (Lucchitta, 1976a, 1976b; Watters, 1988, 1991; Schultz, 2000). The faults responsible for some wrinkle ridges even continue far into the highlands – outside the mare basalts – where they transition into lobate scarps completely independently from volcanism (Howard and Muehlberger, 1973; Watters, 1993; Williams et al., 2019). Recent work has also shown that some wrinkle ridges can also be morphologically very

young (<1 Ga) and far after conclusion of volcanism, indicating a separate source of compressional stress can be responsible for their generation (Williams et al., 2019; Lu et al., 2019; Nypaver and Thomson, 2022). Although I agree that the particular wrinkle ridges discussed in this manuscript are much older and were primarily driven by flexure from mascon and/or mare basalt loading of the lithosphere, when generalizing to other features globally, caution must be exercised when attributing a broader genetic relationship between these morphologic landforms and a volcanism-driven cause.

Reviewer #2 (Remarks to the Author):

Review of: Evidence for Structural Control of Mare Volcanism in Lunar Compressional Tectonic Settings

By: Feng Zhang, Alberto Pizzi, Trishit Ruj, Goro Komatsu, An Yin, Yang Liu, Yongliao Zou

The paper investigates the relation between contractional wrinkle ridges and volcanic landforms in the mare basalts. It is suggested that the paradox of contemporaneous contraction of the mare basalts with continued mare volcanic activity can be explained by strike-slip faults associated with the formation of wrinkle ridges. The authors suggest that strike-slip faults reactivated buried, deeply rooted, basin related normal faults that provided a conduit for magma ascent.

The observation that mare basalt emplacement and contractional deformation are contemporaneous is not new. Evidence of ponded flows on wrinkle ridges was noted by Bryan (1973) and thinning of flows on structural relief as found in ALSE radar sounder data (Phillips et al., 1973) (also see Watters, 1988; Watters and Johnson, 2010). This relation has also been observed in the Columbia River Basalts and the Yakima folds – terrestrial analogs to planetary wrinkle ridges (Reidel, 1984).

It is suggested that basin concentric wrinkle ridge orientations are controlled dominantly by buried normal faults, peak rings, and mechanical properties of the basement. While this may be the case, it is not a necessary condition of the mascon tectonic model. The basin concentric pattern of wrinkle ridges and graben are controlled by stresses from subsidence and flexure of the lithosphere induced by the load of the mare basalts. The volcanic landforms association of wrinkle ridges described in the paper are limited in size and have not been connected to major vents and sources of the flood basalts.

There is an overdependence on the interpretation on strike-slip faults and motion that is not directly observed but interpreted by the orientation of wrinkle ridges. In the primary example, the wrinkle ridge in the Lambert crater area, the authors show ridge segments as two independent ridges; however, they are likely components of a single continuous ridge – two superposed ridges on a common broad arch. Thus, the need to invoke a transfer zone is not obvious. Also, the orientation of this wrinkle ridge is not consistent with localization by a basin related normal fault.

The strike-slip faults associated with planetary wrinkle ridges and the Yakima folds of the Columbia Plateau are accommodation structures (see Watters, *Geology*, 1991). They are likely shallow rooted, probably not throughgoing, and thus not good candidates for magma conduits. I suggest the authors focus more on the thrust faults controlling wrinkle ridges as possible magma conduits. Wrinkle ridge thrust faults are thought to be throughgoing and could communicate with deeply rooted basin normal faults channeling magma. Magma ascent along active thrust faults has been investigated (see Ferre et al., *Int J Earth Sci.*, 2012). Also, evidence of magma ascent along thrust faults has been found on Jupiter's moon, Io (see McGovern et al., *Icarus*, 2016).

Although this is an interesting paper, in its present form I cannot recommend it for publication in *Nature Communications*. After major revisions, the paper has the potential to be a significant contribution to our understanding of the interplay between mare tectonics and volcanism.

Other suggestions are given below.

Line 18, Not sure this should be included as a key point – the relationship has been known since the Apollo era.

Line 35, The point of the last sentence of the abstract is not clear – “more frequent than previously thought?”

Line 40, Delete the word “operation.”

Line 41, Delete the word “whole.”

Line 45 and 49, Consider adding a reference to Watters, JGR, 1988.

Line 51, This statement is misleading - Watters (2022) does not show evidence of this, but predicts it from stress models.

Line 58, This relationship is also observed on the Columbia Plateau. See Reidel (1984).

Line 67, Consider adding a reference to Watters, Geology, 1992.

Line 107, Is there direct evidence of a strike-slip fault in LROC NAC images or is this a notional interpretation?

Line 108, Provide latitude and longitude in text or figure caption.

Line 132, This terminology is confusing - wrinkle ridges are left stepping?

Line 136, Are the authors suggesting that compressional stress largely confined to the mare basalts are effecting buried ancient normal faults and reactivating them as strike-slip faults? This highly speculative.

Line 158, The complex morphology of wrinkle ridge can be accounted for without invoking reactivation of preexisting structures (see Watters, 1988, 1991, 2014, 2022).

Line 165, Again consider adding a reference to Watters, 1992.

Line 175, This is a highly speculative scenario. Why would it be expected that minor strike-slip faults that are essentially accommodation structures confined to the mare basalts coincide and communicate with buried, basin concentric normal faults?

[REDACTED]

Reviewer #3 (Remarks to the Author):

This paper proposes a mechanism to explain the relationship of wrinkle ridges (WR) to volcanic features in the Moon. In consequence, this mechanism would also explain how volcanism can exist in compressional settings. If this mechanism is correct, it would significantly impact our current understanding of the thermal history of the Moon because compression on the Moon would not be due exclusively to cooling after mare emplacement, but compression could instead occur during mare emplacement periods. In general, the promise of the paper is important for the lunar volcanism science community, however, some details need to be improved so the paper can support its results and conclusions. I recommend authors work on the following comments and the paper gets a second round of reviews before it can be published.

Main comments:

1. The description of the mechanism that explains the correlation between WR and volcanic features, lacks clarity.

Lines 111-177 contain the description of this mechanism, however, it is combined with background information and discussion on the validity of the proposed mechanism. With the current arrangement of the text, it is challenging to identify what is the new idea proposed by the authors. I suggest the authors write a specific paragraph to exclusively describe the mechanism, step by step. I suggest avoiding the use of words like "may have" in this description. For example, in line 117, it is written "The central mare load may have exerted tension on the lithosphere, leading to dilation of the basin-concentric faults which provide ideal geological settings favorable for magma ascent." With the current sentence, I cannot assess how confident the authors are about this process occurring, or how important is the role of this process in the proposed mechanism.

2. The visualization of the proposed mechanism lacks details. Figure 3 shows the link between normal faults and strike-slip faults, however, there is no sense of time in this figure. I suggest modifying figure 3 to include a flowchart describing the steps from the creation of the normal faults due to the impact, the change to compressional regime due to the emplacement of the mare basalts... to the formation of volcanic features. Another option is adding numbers to figure 3 to label which process occurred first, and which occurred later.

3. A summary of the information obtained from the other 29 study sites is missing. I suggest the authors summarize, in the main text, the information contained in the supplementary material. I think it is important to state what are the main differences between the observations in the Lambert area and in the other study sites. For example, is the transfer area the same? does the orientation of the WR plays a role in the mechanism described? Is the proposed mechanism supported by all sites?

4. There is no measure of significance of the correlation between the location of volcanic features and WR. While 30 volcanic features coincide with WR, readers cannot assess if this is significant or not. Do most of the WR coincide with volcanic features? Is it just half? If there is no way to assess the significance of this correlation, that should be mentioned in the discussion.

5. The authors should explain with more details what it means, in terms of the thermal history of the Moon, that the WR are produced during mare emplacement and not after cooling of the Moon. With this comment, I want to motivate authors to take the discussion one step further and describe exactly what aspects of the thermal history of the Moon will need to be reviewed considering the results of this paper. Some example questions to discuss: If WR were in fact produced during mare emplacement, then are these features younger than previously thought? What is the percentage of WR that are produced by the cooling of the Moon (if some) compared to the ones produced by mare emplacement?

Specific comments:

6. An explanation on how the transfer zone is delineated is missing or needs additional details.

7. Related to the lack of clarity in the description of the mechanism: How is the low-angle, thrust fault in Figure 3 formed? Is the oblique/slip fault that connects to the normal faults created by the formation of the WRs or is it created by having two sets of WR like in the transfer zone of the Lambert region?

8. If the formation of WR re-activates older normal faults, does it mean the magma within the normal faults had to be warm/buoyant enough to ascend through the newly formed faults? Is there any constraint on how far apart in time a normal fault had to be from the formation of the WR?

9. Lines 121-123 mention that mare emplacement caused compression within the Lambert-SE region. How can mare emplacement cause compression? I understand that a thicker emplacement of mare within the basin center cause a pressure gradient in that direction, but I don't understand why the emplacement result in compression. For compression, two forces with opposite directions need to be collide within the Lambert area.

Reviewer #1 (Remarks to the Author):

This manuscript investigates associations and argues for a link between tectonism and volcanism in a portion of Mare Imbrium and generalizes more broadly to other locations across the lunar maria. Specifically, compressional tectonism represented by wrinkle ridges is proposed to have occurred both contemporaneously with and after volcanism, and that the tectonic structures provided conduits at which magma might ascend. The manuscript also hypothesizes that oblique and/or lateral motion on some faults occurred under the control of pre-existing weaknesses in the lunar crust.

The paper is generally well written, and is an interesting case study with new examples of vents + wrinkle ridges that I had not specifically seen before. However, I do not believe that the broader concepts of association between volcanism and tectonism are particularly new, and any evidence for strike-slip faulting is lacking. [Yes, we agree reviewer's point that lunar wrinkle ridges are tectonic in origin and formed subsequent to (or after) the deposition of their host mare lava flows. This consensus has been reached and accepted by the lunar science community in the past decades. However, just as the reviewer pointed out that the reported cases (vents + wrinkle ridges) in our study have never been concerns before. This study aims to address this newly observed phenomenon by initially and carefully raising a theoretical model for structural control of lunar mare volcanism, which is relevant to (i.e., possibly cooccur with and even after) WR-formation compressional tectonic activities. Based on the detailed investigation of the two focused study cases (Lambert-SE and Herigonius rille regions), we believe that this time we have found solid evidence for potential horizontal movement (lateral slip, appearance resembles strike-slip motions) along the underlying preexisting normal faults. We now focus on the occurrence of deep-rooted crustal fault reactivation within basins accounting for the volcano-tectonic mechanism. We hope that you can find more details in our newly revised manuscript.]

Wrinkle ridges that ponded lava flows and also continued to deform subsequently have previously been documented back to the post-Apollo era (e.g., Bryan, 1973; Schaber, 1973; Schaber et al., 1976). Similar relationships have also been observed with terrestrial tectonism/volcanism in the Columbia River basalts (Reidel, 1984; Watters, 1988). [Thanks for the information! But early proposed volcanic origin for the formation of wrinkle ridges has long ago proved to be not correct by a series of imagery observations and topographic measurements, such as ridge-crater intersections (e.g., Sharpton & Head, 1988) and wrinkle ridges extending into highlands (e.g., Lucchitta, 1976) [also see many early and current works by the team of T. R. Watters (1988, 1991, 1992, 2010, 2014, 2018, 2022)

[REDACTED] They have been widely-accepted to be tectonic deformation (faulting and folding) of mare surface formed subsequent to or a certain time period after ponding of local lava flows (e.g., Yue et al., 2017). Nevertheless, our studied cases indicate that reactivation of underlying normal faults in the basin basement may have potential to provide conduits for magma transport. This means that WR formation-

related compressional tectonics can also trigger later mare eruptions, and thus provide us new implications and insights into the basin thermal evolution of volcano-tectonic activity. Thanks for the mentioned references, which gave us helpful information for understanding the possible mechanism behind this phenomenon. More terrestrial analogs have been added in our newly revised manuscript.]

I find intriguing the hypothesis of a “transfer zone” between adjacent segments of wrinkle ridges as being strike-slip faulting, but no observational evidence supporting strike-slip faults is presented. [We have very carefully re-checked the focused two cases: Lambert-SE and Herigonius rille regions, and have added new updated figures to show surface evidence for potential lateral slip movement at the transfer zones (Please see Figs. 3, S4 and S6 in our supporting information).] Such evidence might include measured lateral displacements across surface features, diagnostic morphology assessment, or numerical modeling of resulting surface landform reliefs/displacements. [The volcano-tectonic interactions on the Moon are always complex and surface features would have been resurfaced, covered, and modified by later eruptions. But, surface features indicative of horizontal tectonic movement can be carefully recognized from a combination of high-resolution images and in particular topography information (Please see Fig. S6).] No distinction is made between discrete faults vs a more distributed shear zone, [We are sorry for not understanding what do the reviewer mean but thanks for the comment. We don't have a clear idea about the discrete faults and distributed shear zones of the review comment. But if the distributed shear zones mean deformation bands, we know some examples from those developing in the sand stones in Utah, which can be compactional or dilational (Okubo et al., *GSA Bulletin*, 2009, p. 474-482; a paper on the concept applied to Mars). The dilatational deformation bands can serve as conduits for fluid of their higher porosity and permeability. As far as we know, strike-slips are brittle and shearing is ductile deformation. So, the processes are completely different.] which one might expect to have significant implications when suggesting these proposed faults as conduits for magma ascent. [Previous studies suggest that under compressional stress, the growth and development of mare wrinkle ridges are often controlled by the underlying preexisting features, such as buried crater rims, pre-existing joint blocks, or linear/ring fractures (e.g., Strom, 1972; Bryan, 1973; Cruikshank et al., 1973; Brace and Kohlstedt, 1980). Since Apollo explorations, one of important remote observations is that the location of wrinkle ridges is always above suggested subsurface basin structures (e.g., Maxwell et al., 1975; Golombek, 1984; Sharpton and Head, 1982), which later is the fundamental for developing the lithospheric-deformation geophysical models to explain a variety of tectonic patterns in mare basins in the past decades of years (e.g., Melosh, 1978; Solomon, 1978; Solomon and Head, 1979, 1980; Freed et al., 2001). Based on these early works, some authors concluded that the location and trend of some wrinkle ridges could be inherited from underlying structures (e.g., Golombek, 1983, 1984). Consequently, the location and trend of some wrinkle ridges (fault may initiate at the mechanical discontinuity contact between the base of the host basalts and basin floor;

Golombek, 1984) may be inherited from the basement-involved normal faults, which reside in the crustal megaregolith (e.g., Richardson and Abramov, 2020). The surface expression of en-echelon folds (e.g., en-echelon pattern mountain-like uplifts with offsets at the transfer zone of Lambert-SE; Figs. 3 and S4) and strike-slip motions in the Herigonius rille region (Figs. S6 and 4C) would be surface manifestations of lateral slip along the underlying preexisting basin-concentric normal faults/fractures, as suggested and argued for strike-slip faults by the early work of Tjia (1970). This is supported by the field study of terrestrial impact craters: some strike slip faults together with a variety of concentric normal faults occurred at the rim zone of the 23-km-diameter Haughton impact structure (Osinski and Spray, 2005; their Fig. 2). Consequently, characterizing the normal fault-ridge intersection zones where eruption centers are located implies that deep-rooted crustal fault reactivation may have provided a valid mechanism for magma transport through fault planes during ridge faulting and folding of basaltic layers.]

Additional Refs:

Brace, W.F. and Kohlstedt, D.L., 1980. Limits on Lithospheric Stress Imposed by Laboratory Experiments. *Journal of Geophysical Research: Solid Earth*, 85: 6248-6252.

Cruikshank, D.P., Hartmann, W.K. and Wood, C.A., 1973. Moon: 'Ghost' craters formed during mare filling. *The moon*, 7(3-4): 440-452.

Golombek, M.P., 1983. Fault type predictions from stress distributions on planetary surfaces: Importance of fault initiation depth, *Lunar and Planetary Science Conference*, pp. 249-250.

Okubo, C.H., Schultz, R.A., Chan, M.A., Komatsu, G. and Team, H.-R.I.S.E., 2009. Deformation band clusters on Mars and implications for subsurface fluid flow. *Geological Society of America Bulletin*, 121(3-4): 474-482.

Richardson, J.E. and Abramov, O., 2020. Modeling the Formation of the Lunar Upper Megaregolith Layer. *The Planetary Science Journal*, 1(2): 1-18.

Sharpton, V.L. and Head III, J.W., 1982. Stratigraphy and structural evolution of southern Mare Serenitatis: A reinterpretation based on Apollo Lunar Sounder Experiment data. *Journal of Geophysical Research: Solid Earth*, 87(B13): 10983-10998.

Wrinkle ridges are complex features that may or may not necessarily include one or multiple narrow crests superimposed atop the structure, which can and often do represent antithetic faults. [We completely agree that antithetic faults along their strike is a typical morphologic characteristic of lunar wrinkle ridges, as well as those on other planetary bodies including our Earth, Mars and Mercury (e.g., Plescia & Golombek, 1986; Schultz, 2000; Andrews-Hanna, 2020).] An example is shown in Figure S5.A, where the main wrinkle ridge crest to the southeast reaches the vent. Notably, the northwest trace as shown in S5.D has higher relief (its hanging wall) on the northeast side as seen in S5.B, indicating it is actually an antithetic fault with opposite vergence to the fault trace drawn on the southeast side of the vent, and with the opposite vergence to what it is shown as having in the figure – meaning it is not the same continuous fault surface expression offset laterally as indicated in S5.C. [Thanks for the reviewer's comments! However, the antithetic thrust faults always reflect near-surface shallow-depth faulted structures

(splays and fold), as indicated by the following Figure 5a directly sourced from Plescia and Golombek (1986). The broad WR arches with uplifted center tops would directly reflect the deeper-rooted thrust faulting activity. The offset, discontinuous arched WRs at the vent place (left WR in Fig. S6) and the one to its east (right WR in Fig. S6) both indicate lateral dislocation landforms. At the same time, note that the shape of the rille segment (nearly linear and parallel to the strike of the en-echelon patterned vents occurring on the left WR) run across the transfer zone should have controlled by the subsurface tectonic features (i.e., faults.) This example could be explained as the antithetic ridge crest forming to the northwest after and not propagating through the vent, without any strike-slip motion required. [According to the sequence of events of our new interpretations, basement-involved normal faults are first to form, and then mare basalt loading and subsequent faulting to form intra-mare thrusts, which intersect with sub-mare normal faults. At the intersection place, horizontal movement along the normal faults occurred, modifying and offsetting the overlying WRs as shown in Figure S6. During the lateral slip motions, the rille-formation flooding event happened and its eruption center is well located at the transfer zone (i.e., the intersection place).] There is also a second more northerly trace not drawn in S5.D but where the zigzagged arrow is covering it up in S5.C, that appears would be continuous with the southeastern fault trace if the vent did not superpose it, further suggesting no lateral/strike-slip motion. [We interpret this as resulting from the boundary sharp ridges formed after the lateral motion to offset the broad arch. For the studied case, differential compressional stresses to form WRs and the stress-caused lateral motions along subsurface normal faults cooccurred at the nearly same time.] Similarly, in Figure 2A, although the narrower superimposed ridges are largely absent in the transfer zone, the broader arched surface expression of the wrinkle ridge is still present throughout the transfer zone connecting the two segments, suggesting no large-scale changes in the deeper structure (other than it simply being a non-planar fault) that would require distinct strike-slip faulting. [In the transfer zone of Lamber-SE, they are not broader arched surface expression of the typical WRs but mountain-like hills with steep-slope SW flank and gentle-slope NE flank, also with flow lobe-like features, please see the added Fig. S4 in our supporting information.]

[REDACTED]

Another consideration not mentioned in this paper is that although wrinkle ridges occur in the lunar mare and other planetary basaltic plains, this surface morphologic classification does not necessarily require its underlying faults be controlled by or have a driving force from volcanism itself. Rather, wrinkle ridges are likely a form of surface expression resulting from compressional faulting and folding (possibly but not necessarily driven by volcanic emplacement) in a layered material consistent with what layered flood volcanism presents (Lucchitta, 1976a, 1976b; Watters, 1988, 1991; Schultz, 2000). The faults responsible for some wrinkle ridges even continue far into the highlands – outside the mare basalts – where they transition into lobate scarps completely independently from volcanism (Howard and Muehlberger, 1973; Watters, 1993; Williams et al., 2019). [We totally agree the reviewer’s point. Considering that nearly all wrinkle ridges are spatially limited in the basin interiors and near the basin-margins (Thompson et al., LPSC abstract 2782, 2018), their areal distribution and together with basin-margin graben features and locations form the basis for the built-up of the mascon tectonic models (Melosh, 1978; Freed, et al., 2001). Current gravitational observations also indicate that the wrinkle ridges in the roughly circular mares are consistent with the previous mascon tectonic models, and in non-mascon mares, wrinkle ridges are found to relate to the shape and geometry of regional-scale lowlands the mare basalts occupy (Thompson et al., 2018). Therefore, and generally speaking, the formation of lunar wrinkle ridges is mainly driven by basin-related isostatic adjustment due to mare filling, loading, and cooling, though the global cooling of the Moon dominates a longer time to continue exerting a certain influence on the WR-growth and development. But we believe that this influence caused by global cooling should be very limited in scales, or else, there would be more WRs in the highlands up to date. Obviously, it is not the case.] Recent work has also shown that some wrinkle ridges can also be morphologically very young (<1 Ga) and far after conclusion of volcanism, indicating a separate source of compressional stress can be responsible for their generation (Williams et al., 2019; Lu et al., 2019; Nypaver and Thomson, 2022). [We also agree this. These interpreted very young WRs are also located in mares. But, by comparison, global contraction on the Moon is estimated to be an order of magnitude smaller than on Mercury, and this suggests that the smaller scale of lunar wrinkle ridges most likely form primarily by load-induced subsidence of the mare basalts (Schleicher et al., Icarus, 2019). This is also evidenced from the observation that the wrinkle ridges are very rare to be seen in the highlands (only limited to basin-highland transition zones). These works that interpreted very young wrinkle ridges in the lunar maria explain them as resulting from global thermal contraction (recent cooling and possible orbital recession), a very different situation from our cases, which occurred in the period of 3.0-4.0 Ga ago with the highest eruption flux period throughout the whole lunar history.] Although I agree that the particular wrinkle ridges discussed in this manuscript are much older and were primarily driven by flexure from mascon and/or mare basalt loading of the lithosphere, when generalizing to other features globally, [Thanks for recognizing this point.] caution must be exercised when attributing a broader genetic relationship between

these morphologic landforms and a volcanism-driven cause. [Thanks for pointing out this. We have taken very care to relate WR-formation tectonics to volcanism. However, we here must declare a possible misunderstanding that our interpretations do not mean that these morphologic landforms are driven by volcanism. Actually, and strictly, our general idea is to conclude that a link of the intra-mare WR-formation compressional tectonic-caused fractures of weakness (due to lateral motions) and reactivation of basin basement-involved normal faults may provide viable conduits for magma transport, and hence erupt at the normal fault-WR intersection place.]

Reviewer #2 (Remarks to the Author):

Review of: Evidence for Structural Control of Mare Volcanism in Lunar Compressional Tectonic Settings

By: Feng Zhang, Alberto Pizzi, Trishit Ruj, Goro Komatsu, An Yin, Yang Liu, Yongliao Zou

The paper investigates the relation between contractional wrinkle ridges and volcanic landforms in the mare basalts. It is suggested that the paradox of contemporaneous contraction of the mare basalts with continued mare volcanic activity can be explained by strike-slip faults associated with the formation of wrinkle ridges. The authors suggest that strike-slip faults reactivated buried, deeply rooted, basin related normal faults that provided a conduit for magma ascent.

The observation that mare basalt emplacement and contractional deformation are contemporaneous is not new. Evidence of ponded flows on wrinkle ridges was noted by Bryan (1973) and thinning of flows on structural relief as found in ALSE radar sounder data (Phillips et al., 1973) (also see Watters, 1988; Watters and Johnson, 2010). [Thanks for providing these useful references. We agree that mare basalt emplacement and contractional deformation show signs of a complex relationship, in particular these volcanic eruption vents on WRs reported in our study. But, the potential close relationship between WR-formation compressional tectonic-volcanism interactions is yet unclear and the operational mechanism behind the observation is still very poorly constrained.] This relation has also been observed in the Columbia River Basalts and the Yakima folds – terrestrial analogs to planetary wrinkle ridges (Reidel, 1984). [Thanks for the information, they are very good terrestrial analogues that can serve as direct evidence for mafic eruptions occurring in compressional tectonic settings, in particular associated with contractional deformation in the basaltic Columbia plateau. This important reference is now cited in our new-version manuscript. In addition, more terrestrial analogue studies have been added to illustrate that reactivation of preexisting deep-rooted crustal normal faults may have provided a valid mechanism for magma transport through fault planes during ridge faulting and folding of basaltic layers.]

It is suggested that basin concentric wrinkle ridge orientations are controlled dominantly by buried normal faults, peak rings, and mechanical properties of the basement. While this may be the case, it is not a necessary condition of the mascon tectonic model. [We agree the reviewer's point. But, we never said that the basin concentric faults dominantly controlled the orientation of the WRs, but should have exerted a certain influence on WRs because their patterns are always affected by the underlying preexisting topography. The Lambert-SE and Herigonius-rille volcanic regions are the two mainly focused cases of interaction between the two sets of discontinuity zones (i.e., the intersection place between mare-hosting WRs and sub-mare basin-concentric normal faults). Both are located within the influenced domain (i.e., basin ring/rim zones) of basin mascon tectonics.] The basin concentric pattern of wrinkle ridges and graben are controlled by stresses from subsidence and flexure of the lithosphere induced by the load of the mare basalts. [We completely agree with the reviewer's point. The complex patterns of wrinkle ridges are now explained as resulting from a consequence of several reasons: impact mascon tectonics, the load of mare basalts, sub-mare topography, intrusion stress tectonics, and so on. However, most documented cases in our study are well located in the regions at or near basin ring/rim zones, and thus, we interpret their formation and development to have been significantly influenced by basin-scale tectonic structures.] The volcanic landforms association of wrinkle ridges described in the paper are limited in size and have not been connected to major vents and sources of the flood basalts. [Wrinkle-ridge pattern-associated mare volcanism described in our work includes varying-size volcanic edifices from small shield volcanoes (with relatively low effusion rates) to rille-formation flooding basalts with high effusion rates and long-lived (months to years) eruptions (e.g., Wilson and Head, 1981; Head and Wilson, 1992; Williams et al., 2000; Hurwitz et al., 2012). For example, in the Lambert-SE region, the volcanic shield has a summit crater of ~3 km in diameter and also a sinuous rille sourcing from the shield flank all indicating that the Lambert-ridge associated volcanism is relatively large in scale. In addition, the Herigonius rille (Fig. S6 in our supporting information) and some other rille-formation flooding events (e.g., Figs. S8, S9, S13, S14, S17, S19, and S21) are all related to eruptions with volume eruption rates in the range between 10^4 - 10^6 m³s⁻¹ estimated by geophysical models (e.g., Wilson and Head, 2017; Head and Wilson, 2017). Based on the channel volume, sinuous rilles are modeled to be fed by eruption activity with a duration of 100-300 days and erupted volumes of ~100 to 2000 km³, implying mantle partial melt source regions of up to ~35 km in vertical extent (Wilson and Head, 2017). Additionally, it should be noted that, though volcanic domes and cones are small shields formed from low effusion-rate flow accumulation, they are just terminal products of one complete volcanic event (with sequential eruption phases) that always began with explosive and effusive eruptions of high-volume eruption rates to form rilles' head depressions and sinuous rilles (Wilson and Head, 2018). Thus, the volcanic landforms associated with wrinkle ridges are not limited in size but with some feeding the surrounding mare plains, as inferred from lengths of sinuous rilles cutting across different mare units. Obviously, their basaltic

nature and scales indicate that some are connected to major vents and sources of the flood basalts.]

Additional Refs:

Hurwitz, D.M., et al., 2012. Origin of lunar sinuous rilles: Modeling effects of gravity, surface slope and lava composition on erosion rates during the formation of Rima Prinz. *J. Geophys. Res. Planets* 117, E00H14.

Williams, D.A., et al., 2000. A reassessment of the emplacement and erosional potential of turbulent, low-viscosity lavas on the Moon. *J. Geophys. Res. Planets* 105, 20189–20205.

Wilson, L., Head, J.W., 1981. Ascent and eruption of basaltic magma on the Earth and Moon. *J. Geophys. Res.* 86, 2971–3001.

Wilson, L., Head, J., 2018. Controls on lunar basaltic volcanic eruption structure and morphology: Gas release patterns in sequential eruption phases. *Geophys. Res. Lett.*, 45(12): 5852-5859.

There is an overdependence on the interpretation on strike-slip faults and motion that is not directly observed but interpreted by the orientation of wrinkle ridges. [Indeed, to the lunar science community, this once was, is now, and will be a long-standing debate since Apollo exploration. Given that the wrinkle ridge pattern is commonly controlled and guided by sub-mare topography and structures, using their surface expressions to trace basement fault directions and patterns may be an effective and practical solution to address this issue currently. However, the two cases of Lambert-SE and Herigonius rille documented in this study may provide relatively much stronger evidence to support this hypothesis. We believe that you will find more details by going through our newly revised manuscript.] In the primary example, the wrinkle ridge in the Lambert crater area, the authors show ridge segments as two independent ridges; however, they are likely components of a single continuous ridge – two superposed ridges on a common broad arch. Thus, the need to invoke a transfer zone is not obvious. [According to the common lunar WR characteristics (broad arch with superposed and boundary sharp ridges/scarps), we first mapped these ridge segments, as shown in Fig. 1 below. The two groups of ridge segments, marked in yellow and light-blue dashed lines respectively, clearly can be divided into two independent groups connected by a transfer zone. The transfer zone shows very different features distinct from those characterizing wrinkle ridges (Fig. 2). For example, in Figure 2 (Kaguya TC map, ~10 m/pixel) below, lobate flow front/scarp-like features can be seen in several locations at the foot of the gentle-slope NE flanks of the mountain-like hills in the transfer zone. If you check them in high-resolution NAC images, these features will be clearer. Further, it seems that in the transfer zone, the two independent ridges are linked by three segmented offset hills, which are spatially distributed in *en-echelon* pattern. Clearly, these hills are very different from the narrow wrinkle ridges shown in Figs. 1 and 2 (yellow and light-blue dashed lines). The shapes and outlines of these hills can be directly checked from the color SLDEM2015 map (Baker et al., 2015) and the SLDEM-derived slope azimuth map (Fig. 3, available by using webgis tool: QuickMap). These hills have a steep-slope SW flank and a gentle-slope NE flank, significantly distinct from those typical WRs in the studied

region. The above-mentioned observations give us the confidence to define the transfer zone. This type of mountain belt or fault block mountains was used by Tjia (1970) as fault traces with offset mountains (drag folds) indicative of subsurface strike-slip faulting in the substratum.] Also, the orientation of this wrinkle ridge is not consistent with localization by a basin related normal fault. [The alignment of these segmented hills in the transfer zone forms a line concentric to the basin center, and also nearly parallel to the nearby basin ring shown in Figs 2 and 4B (black dashed lines) in our manuscript. This is also clearly marked in our update Figure 4B with a broad grey arrow pointing to the Imbrium basin center. Thus, the location of the transfer zone is possibly above the basin ring-zone normal faults rooted in the basement substrate. Actually, in our study, the transfer zone is defined as the intersection or overlap place between WRs and predicted underlying basement-involved normal faults of weakness.]

Figure 1. The two independent ridges in the Lambert-SE area are defined according to the common lunar wrinkle ridge characteristics with broad arch superposed by ridges and boundary secondary ridges at its both sides. The yellow-color ridge segments and the light-blue-color ones can be clearly divided into two distributed groups (i.e., two independent major WRs). The detailed geology of the transfer zone can be seen in Fig. 2 below.

Figure 2. In the transfer zone, basaltic flow deposits and lobate-like flow fronts/scarps are observed. The three mountain-like structures (white dashed ellipses and blue lines) are more like *en-echelon* hills rather than narrow, sharp ridge segments as the one shown in the top of the figure. The outline of these hills can be clearly seen from DEM-based topography in Fig. 3 (slope azimuth map) and Fig. 4 (colored SLDEM2015 map).

Figure 3. SLDEM2015-derived slope azimuth map for the three *en echelon* arrangement hills in the defined transfer zone. The map can also be accessible by using webgis tool: QuickMap (<https://quickmap.lroc.asu.edu/>).

Figure 4. Colored SLDEM2015 map showing topographic variation for the three *en-echelon* arrangement hills in the transfer zone. These hills have a steep-slope SW flank and a gentle-slope NE flank, significantly distinct from typical WRs.

The strike-slip faults associated with planetary wrinkle ridges and the Yakima folds of the Columbia Plateau are accommodation structures (see Watters, Geology, 1991). They are likely shallow rooted, probably not throughgoing, and thus not good candidates for magma conduits. [Yes, we agree with the reviewer's point. Wrinkle ridge-related strike-slip faults are mainly shallow-rooted in intra-mare, but it will tell a different story if the WR-related strike-slip fault is located above or its part directly linked to the deep-rooted basin ring/rim normal faults. Our reported cases clearly appear to support this idea. We now use a new tectonic scenario of reactivation of preexisting crustal normal faults to explain this mechanism.] I suggest the authors focus more on the thrust faults controlling wrinkle ridges as possible magma conduits. [Thanks for the suggestion! The newly revised manuscript is now more focused on the reactivation during compression of the deep-rooted preexisting normal fault with oblique-slip kinematics, as expected at the transfer zones. At shallow crustal level, deformation is partitioned between new formed low-angle thrusts (and folding) and oblique/strike-slip faulting, commonly along the pre-existing discontinuities (i.e. strain partitioning). The overall high-angle oblique/strike-slip and inverted normal fault structures might have served as conduits for mantle magma ascent and eruption.] Wrinkle ridge thrust faults are thought to be throughgoing and could communicate with deeply rooted basin normal faults channeling magma. Magma ascent along active thrust faults has been investigated (see Ferre et al., Int J Earth Sci., 2012). Also, evidence of magma ascent along thrust faults has been found on Jupiter's moon, Io (see McGovern et al., Icarus, 2016). [Many thanks for the useful information from previous studies. These works are now cited in our revised manuscript. These inspirational cases are good analogues for supporting

how active contractional tectonics controls the transport of magma (also telling us that the lower viscosity magma is more likely to occur), in particular under the control of pre-existing structures (i.e., the architecture of pre-mare faults and fractures representing as weaknesses in rocks).]

Although this is an interesting paper, in its present form I cannot recommend it for publication in Nature Communications. After major revisions, the paper has the potential to be a significant contribution to our understanding of the interplay between mare tectonics and volcanism. [Many thanks for recognizing the scientific value of our manuscript. With your instruction and suggestions, we hope that our newly revised manuscript has been greatly improved.]

Other suggestions are given below.

Line 18, Not sure this should be included as a key point – the relationship has been known since the Apollo era.

This key point of the relationship has been removed.

Line 35, The point of the last sentence of the abstract is not clear – “more frequent than previously thought?”

The sentence is now deleted in our revised manuscript. However, as the cases reported in this study (we believe that more will be found in future), if the formation of wrinkle ridges concurrent with mare volcanism (i.e., WR-formation compressional tectonic-caused eruptions), and thus, volcanism in zones of active compression on the Moon may be more frequent than previously thought.

Line 40, Delete the word “operation.”

Deleted as suggested.

Line 41, Delete the word “whole.”

Deleted as suggested.

Line 45 and 49, Consider adding a reference to Watters, JGR, 1988.

Thanks for mentioning the important work, and now added as suggested (ref. 1).

Line 51, This statement is misleading - Watters (2022) does not show evidence of this, but predicts it from stress models.

Thanks for pointing this. Now, the sentence is updated with beginning by “Stress models from the load on lunar lithosphere⁷ suggest that

Line 58, This relationship is also observed on the Columbia Plateau. See Reidel (1984).

This reference to Reidel (1984) is now added into the sentence where terrestrial examples are mentioned (ref. 23).

Line 67, Consider adding a reference to Watters, *Geology*, 1992.

Thanks for the information. The reference to Watters (1992) is now added (ref. 27).

Line 107, Is there direct evidence of a strike-slip fault in LROC NAC images or is this a notional interpretation?

As for the Lambert-SE case, the possible existence of a subsurface strike-slip fault in the transfer zone is mainly inferred and predicted from the morphological expressions of WR patterns (also see our replies to comments somewhere else above). We should also note that thrust deformation was partly coeval with volcanism (e.g., Reidel, 1984; Galland et al., 2007), thus, finding direct evidence for strike-slip faults by only using remote images would be difficult to realize due to syn- and post-tectonic activities, such as erosion and coverage by post-tectonic hot lava, impact, mass wasting, and regolith formation. However, the direct evidence of a strike-slip fault (or lateral slip motions) is from the case of the Herigonius rille area in southern Procellarum (Please see our new Figure S6 in the supporting information, and also attached as Figure 5 below). We found that the Herigonius rille-hosting WR and another WR to its east both appear to have been cut off by strike slip-like horizontal movement along the SW-NE direction, respectively with a certain distance offset for both major WR segments. If using a line to link them, the line is nearly concentric to the Humor basin center. From NW to SE, the two major NW-SE-trending WRs have been moving right with a clear offset at the WR-rille intersection zone, maybe showing a sign of subsurface strike slip-like horizontal movement. We have also added a sketch map for the Herigonius case in Figure 4C.

Figure 5. Kaguya TC evening Map (top) and SLDEM2015 data (bottom) for the Herigonius rille (lava channel) in southeastern Procellarum (Location: 12.5052°S, 37.4328°W) with two elongated head depressions cutting across a wrinkle ridge (WR). (top) The two rille-head vents of elongation in shape are arranged in an en-echelon pattern and oriented along the SW-NE direction. (Bottom) SLDEM2015 data is used to show the regional topographic variation for defining the outline of broad anticlinal arches of the major WR segments. The black dashed lines illustrate the boundaries of WRs. Clearly, the two major WRs both are cut off by rilles with a certain distance offset for their two segments, respectively. Further, this is to show the scenario for the possible strike slip-like horizontal movement along the preexisting structural weakness, e.g., basin-ring normal faults in the basement associated with the Humorum basin formation and loading tectonics.

Line 108, Provide latitude and longitude in text or figure caption.

The latitude and longitude are now provided in the Figure 3 caption.

Line 132, This terminology is confusing - wrinkle ridges are left stepping?

The term “left stepping” is commonly used in structural geology to "specify" the spatial

arrangement of a structure with an en echelon pattern (folds, normal faults, thrusts, etc). Left-stepping refers to the arrangement in which one fault segment occurs to the left of the adjacent segment from which it is being viewed. The contrary is right-stepping (see for ex. page 176 from lecture by Ben van der Pluijm: <https://www.files.ethz.ch/structuralgeology/jpb/files/english/5wrench.pdf>). Actually, this arrangement is not only of "descriptive" use, but it is very important in kinematic terms because it assumes a particular relationship with the stress field, and in the specific case with the horizontal sigma max (which would be opposite if we analyzed *en-echelon* normal faults with respect to a folds or thrusts in echelon). In practice if the *en-echelon* structures forming the left stepping are folds or thrusts, they agree with a left shear, which is the one expected for faults like the Lambert-SE that are oriented about NNW-SSE (Please see Figures 2-4 above and Figure 6 below).

Figure 6

Line 136, Are the authors suggesting that compressional stress largely confined to the mare basalts are effecting buried ancient normal faults and reactivating them as strike-slip faults? This highly speculative.

A recent study by García et al. (2019) propose a model that seems to us quite similar (also in terms of structure orientation) to ours for the formation of *en-echelon* folds in the Andes chain following the reactivation in compression of pre-existing normal faults. We here only copy their point (2) in the “Conclusions” part as below:

- (2) We propose that the surface expression of the Lomas de Carabajal structures (mainly NNW-trending, left stepping *en-échelon* folds) is controlled by a local left-lateral transpressive positive palm-tree structure associated with a partially inverted Cretaceous normal fault. In this scenario ENE-WSW orientred Andean compression acting upon acting the NW-trending northeastern border of the Alemanía sub-basin results in transpression with a sinistral strike-slip component that causes the *en-échelon* arrangement of the folds.
- (3) Four late Pleistocene to Holocene terrace surfaces at progressively

Again, it seems to us that there is now evidence in many multi-deformed orogenic belts everywhere, which underlines the importance of tectonic reactivation/inversion of pre-existing structures, particularly in basin settings (also see more references of

terrestrial analogues in our newly revised manuscript; e.g., refs. 62-64, 66, and 67.). An average basalt thickness of 0.74 km in lunar mare basins has been derived from gravity and topographic estimation (Gong et al., 2016). The basalt loading last a long time periods from the basin formation (most > 3.8 Ga) to ~3.0 billion years ago, the duration of which depends on each basin's thermal evolution history. Long-time gravitational equalization between the basalt capping layer and basement and the magma chamber inflation in the upper mantle (or intrusion) may help the pre-mare normal faults to propagate to the surface by linking and/or being reactivated during contractional tectonic activity. In particular, if the compression is continued that may affect in their reactivation or movement along their preexisting trend.

The important reference is also provided here:

García, V. H., et al., 2019. Late Quaternary tectonics controlled by fault reactivation. Insights from a local transpressional system in the intermontane Lerma valley, Cordillera Oriental, NW Argentina. *Journal of Structural Geology*, 128, 103875.

Line 158, The complex morphology of wrinkle ridge can be accounted for without invoking reactivation of preexisting structures (see Watters, 1988, 1991, 2014, 2022).

Thanks for pointing out this. We have specified the sentence with the beginning by "Under some circumstances, these complex WRs geometric patterns may also reflect...".

Line 165, Again consider adding a reference to Watters, 1992.

The reference to Watters (1992) is now added (ref. 27).

Line 175, This is a highly speculative scenario. Why would it be expected that minor strike-slip faults that are essentially accommodation structures confined to the mare basalts coincide and communicate with buried, basin concentric normal faults?

The 'strike-slip faults' are now not our focus, but we consider them as possible structures formed during the WR-forming tectonic folding and faulting activities. We found that the positions of most volcanic eruptive centers are linked to wrinkle ridge-forming compressional tectonics acting above basin basement-involved ring/rim normal faults. Characterizing the normal fault-ridge intersection zones where eruption centers are located implies that deep-rooted crustal fault reactivation may have provided a valid mechanism for magma transport through fault planes during ridge faulting and folding of basaltic layers.

[REDACTED]

Reviewer #3 (Remarks to the Author):

This paper proposes a mechanism to explain the relationship of wrinkle ridges (WR) to volcanic features in the Moon. In consequence, this mechanism would also explain how volcanism can exist in compressional settings. If this mechanism is correct, it would significantly impact our current understanding of the thermal history of the Moon because compression on the Moon would not be due exclusively to cooling after mare emplacement, but compression could instead occur during mare emplacement periods. In general, the promise of the paper is important for the lunar volcanism science community, however, some details need to be improved so the paper can support its results and conclusions. I recommend authors work on the following comments and the paper gets a second round of reviews before it can be published.

Main comments:

1. The description of the mechanism that explains the correlation between WR and volcanic features, lacks clarity. [This main text of this content has been reworked and its clarity, we believe, is now greatly improved.]

Lines 111-177 contain the description of this mechanism, however, it is combined with background information and discussion on the validity of the proposed mechanism. With the current arrangement of the text, it is challenging to identify what is the new idea proposed by the authors. I suggest the authors write a specific paragraph to exclusively describe the mechanism, step by step.

[The sequence of all the events of the whole story that we want to tell is summarized below: (1) Impact excavation and basin formation; (2) Isostatic central uplift (mascon formation and relaxation) soon after impact and the transient crater collapses to form ring normal faults, which cut through the crust with their dip angle (i.e., angle with respect to the surface) gradually decreasing in the mantle (see Johnson et al., Science, 2016); (3) After a long period of basin infilling (lava accumulation in the lower topography of the basin center at the early stage), elastic failure due to loading accounts for the concentric normal faults and their dilation, resembling narrow buried lava-flooded rift valleys in the underlying feldspathic/noritic crust (Andrews-Hanna et al., 2014), which is caused by central loading and resulting subsidence; (4) With time, the locations of volcanic eruption centers shift outward and magma transport is mainly controlled by extensional structures (e.g., faults) in the ring/rim zone (e.g., Schaber, 1973; Schaber et al., 1976; Solomon, 1978), consistent with the loading adjustment and local overcoming of global compressive stresses (Solomon and Head, 1979; Head and Wilson, 1992); (5) Many tectonic features formed through flexure under the resulting mare loads and interior cooling (Solomon and Head, 1980), including extensional structures graben with a distance away outward from the basin rim and contractional wrinkle ridges in the interior; (6) As compressional stress, with time, became the prevailing controlling force to dominate the evolution of the Moon, more thrusts and folds of mare basalts began to form, but many are under the structural control of pre-

existing topography (e.g., underlying normal faults and fractures in the basement); (7) Slip and reactivation of sub-mare crustal normal faults under a dominant contractional stress field may have occurred, hence modifying the strike and shape of WRs (e.g., the Lambert-SE and Herigonius rille cases in Figs. 2, 4, S6, and other cases in our supporting information); (8) Surface volcanic eruptions occurred through the faults and even extensional features formed during the faulting and folding of basaltic layers (e.g., the Lambert-SE and Herigonius transfer zones defined in our study; the relationship between lateral slip versus opening along fault segments). In general, the abovementioned processes from (1) to (6) have been confirmed in previous works, while the processes involved in (7) and (8) are the new idea proposed by our study. The related text has been also rearranged and improved accordingly, please see lines “160-203” and “251-316” for more details.] I suggest avoiding the use of words like “may have” in this description. For example, in line 117, it is written “The central mare load may have exerted tension on the lithosphere, leading to dilation of the basin-concentric faults which provide ideal geological settings favorable for magma ascent.” With the current sentence, I cannot assess how confident the authors are about this process occurring, or how important is the role of this process in the proposed mechanism. [The dilation of concentric normal faults may favor magma ascent taking the form of dikes in the crust, considering that dikes should have very high volumes, comparable to the volumes associated with many observed flows and sinuous rilles (see Head and Wilson (1992) for the detail). In addition, dilation of concentric normal faults may allow the occurrence of horizontal movement along the faults more likely. Basin interior loading and mascon relaxation will result in the central subsidence producing oblique dilation (also, resembling the formation of basin ring graben caused by central load) due to exerting radiating tensile normal stresses on the concentric normal faults (Schultz and Zuber, 1994), and at the same time, volcanic activity implies that abundant dikes reside in the crust (Head and Wilson, 1992, 2017) and thus, magma intrusion into the normal faults will also induce extension formation or dilation of the intruded normal faults.]

Additional Refs:

Schaber, G.G., 1973. Lava flows in Mare Imbium: Geologic evaluation from Apollo orbital photography. Lunar and Planetary Science Conference, 4: 73-92.

Schaber, G.G., et al., 1976. The scarcity of mappable flow lobes on the lunar maria: Unique morphology of the Imbrium flows. Lunar and Planetary Science Conference, 7: 2783-2800.

Schultz, R. and Zuber, M., 1994. Observations, models, and mechanisms of failure of surface rocks surrounding planetary surface loads. Journal of Geophysical Research: Planets, 99(E7): 14691-14702.

2. The visualization of the proposed mechanism lacks details. Figure 3 shows the link between normal faults and strike-slip faults, however, there is no sense of time in this figure. I suggest modifying figure 3 to include a flowchart describing the steps from the creation of the normal faults due to the impact, the change to compressional regime due to the emplacement of the mare

basalts... to the formation of volcanic features. Another option is adding numbers to figure 3 to label which process occurred first, and which occurred later. [Thanks for the useful suggestion! A simplified flowchart is now added to the right of Fig. 3 to show the detail.]

3. A summary of the information obtained from the other 29 study sites is missing. I suggest the authors summarize, in the main text, the information contained in the supplementary material. [Thanks for pointing out this point. The summary is now added in the main text. Please see lines 204-249 in our revised manuscript.] I think it is important to state what are the main differences between the observations in the Lambert area and in the other study sites. For example, is the transfer area the same? [We only defined the eruption centers at the Lambert-SE and Herigonius rille regions as transfer zones because strong evidence (see Figs. 3, S4 and S6, and 4) supports lateral slip movement along the possible sub-mare normal faults at the basin ring/rim zones. This has been specified in the main text, lines 222-224.] does the orientation of the WR plays a role in the mechanism described? [The orientation of the WRs should have played an important role in the mechanism described. The differential stresses caused by compressional tectonics parallel to or with an angle to the subsurface normal faults will produce different effects. For example, the crosscutting relationship between WRs and ring normal faults in the Lambert-SE and Hegionus regions are favorable places for causing lateral slip movement along the normal faults (e.g., early extensional movement in the direction radiating to the basin center, and later compressional tectonics with a direction concentric to the basin center).] Is the proposed mechanism supported by all sites? [The tectonic inversion of inherited normal faults oriented obliquely with respect to the direction of the maximum horizontal stress axis (σ_1) as in the case of the Lambert SE transfer zone, may be a rather widespread phenomenon on the Moon, as it is also supported by other examples shown in the Supplementary information (e.g., Figs. S6-S22). Analogue modelling and field examples on Earth, indeed, show that oblique compression favors oblique-slip reactivation of normal faults. We have divided these sites into four classes (See lines 204-249). The former three classes support our mechanism, while the fourth class does not show such a clear relationship with large-scale basin/crater ring/rim structures. We keep them to be further investigated and solved through future efforts. However, they clearly show a relationship with WRs.]

4. There is no measure of significance of the correlation between the location of volcanic features and WR. [Sorry for the confusion. We have divided these targets into four classes: **type 1** like the Lambert and Herigonius cases mainly discussed in the main text with volcanic features located in the intersection place between WRs and subsurface basin-concentric ring normal faults, where horizontal movement along the normal faults is interpreted to have occurred; **type 2** points to the cases with their eruption centers located at the intersection zones between WRs and subsurface faults/fractures (Figs. S7-S12), but no clear evidence for the horizontal movement along faults/fractures is observed; **type 3** includes cases where mare volcanism occurred on

the broad arch of WRs or directly served as fissure-like segments of WRs (see Fig. S14). These eruption center-hosting WRs are concentric to nearly circular basins or buried impact structures, and located at the basin/crater ring/rim zones where normal faults and extensional fractures preferentially tend to occur (Figs. 13-22); **Type 4** contains mare eruption centers which are spatially correlated with WRs. However, the relationship is not so clear as those of types 1 to 3. We leave the type 4 further to be investigated in our future efforts.] While 30 volcanic features coincide with WR, readers cannot assess if this is significant or not. [The context-mentioned volcanic features include varying-size shields (mare domes and cones), large-scale fissure-like and sinuous rille-formation eruption centers, and any combination of them. Their locations are summarized: the intersection zone between WRs and inferred basin-ring normal faults/fractures where WRs are twisted off and deformed and hence horizontal movement along normal faults may have occurred; the volcanic vents formed in the normal fault-WR intersection region but no observable horizontal movement occurred (one possible explanation is that surface clues for lateral slip has been removed by the emplacement of hot lava); the eruption centers are situated on or along the broad arch of WRs, which are located at the basin/crater ring/rim zones and concentric to the basin/crater center; and those are located in the complex WR-pattern zone but do not show the clear relationship as indicated by the other three categories mentioned above.] Do most of the WR coincide with volcanic features? [Yes, most of them.] Is it just half? [Generally speaking, among 30 of all, at least 24 show clear coincidence with volcanic features (Lambert, Herigonius, Figs. S7-S23). We believe that more cases will be identified and are waiting for being investigated in detail in future efforts.] If there is no way to assess the significance of this correlation, that should be mentioned in the discussion. [This issue has been addressed, as the reviewer suggested. Please read our revised manuscript for more details. In particular, the relevant descriptions and discussion are contained in Sections “Global distribution” and “Tectonic inversion origin of the lunar wrinkle ridges”]

5. The authors should explain with more details what it means, in terms of the thermal history of the Moon, that the WR are produced during mare emplacement and not after cooling of the Moon. [Sorry for the misunderstanding. We never said that the WRs are produced during mare emplacement and not after cooling of the Moon. Our new idea is to show during the post-impact basin evolution, compressional tectonics after mare emplacement and with cooling can trigger volcanic eruptions by fully fault reactivation, which cause surface deformation and folding to form WRs, under a dominant contractional stress field. Under such conditions, high-angle oblique/strike-slip faulting and associated extension along releasing zones assisted by bending-movement normal faulting at the WR-crest coeval and stress-compatible with the contractional wrinkle ridge tectonics appear to provide an efficient pathway for ascending magma. Thus, volcanic activity overlapped the periods of wrinkle ridge growth and development. Thus, the formation of some contractional wrinkle ridges concurrent with mare volcanism. Gravity-caused tectonic adjustment due to mass loading may have been the dominant factor to control the evolution of the

basin tectonics and mare fillings. That may be the reason why the nearly all wrinkle ridges are areally distributed in the lunar maria (i.e., the basin interior) (Thompson et al., 2018), only with very few extending into the nearby highland regions (e.g., Maxwell et al., 1975; Plescia and Golombek, 1986; Wilhelms, 1987). The detailed description has been added in Section “Constraints on lunar thermal evolution models”.] With this comment, I want to motivate authors to take the discussion one step further and describe exactly what aspects of the thermal history of the Moon will need to be reviewed considering the results of this paper. [Based on our results and analysis, it can be predicted that some WRs and resulting eruptions should have occurred simultaneously, and thus many mare eruptions should postdate their host WRs and those (WRs) nearby. The previous reached consensus that all the mare wrinkle ridges should postdate their host mare units should be reviewed considering that WR-formation compressional tectonics can also trigger later mare eruptions. Thanks for the reviewer’s suggestion, we have added more discussion in our newly revised manuscript. Please see the Section ‘Constraints on lunar thermal evolution models’ in “Discussion” part.] Some example questions to discuss: If WR were in fact produced during mare emplacement, then are these features younger than previously thought? [WR formation should overlap with the time periods of mare eruptions. This means that either one or any combination of pre-, post-, and syn-mare WRs would have occurred.] What is the percentage of WR that are produced by the cooling of the Moon (if some) compared to the ones produced by mare emplacement? [This is a good question, but also a very big one that is difficult to address at the moment. But, these WRs formed concurrent with mare emplacement should tend to locate at the basin (mascon) center-outside zones with distribution of sub-mare normal faults or fractures/joints. A prerequisite for comparing them is that two kinds of WRs (formed during mare emplacement and the cooling of the Moon) must be confidently identified and carry out global search. This is a challenging but very interesting task that can be considered in future.]

Specific comments:

6. An explanation on how the transfer zone is delineated is missing or needs additional details.

[We here define the transfer zone as the intersection place (with volcanic eruptive centers) between WRs and inferred basin ring/rim normal faults where surface expressions for possible occurrence of differential compression stress-caused horizontal movement along the faults are observed. At current stage, only the main text-focused Lambert-SE and Herigonus cases satisfy this condition. We have specified this point in the main text. Please see lines 222-224.]

7. Related to the lack of clarity in the description of the mechanism: How is the low-angle, thrust fault in Figure 3 formed? [Under the conditions described in this study, high-angle oblique/strike-slip faulting and associated extension along releasing zones assisted by bending-movement normal faulting at the WR-crest coeval and stress-compatible with the contractional wrinkle ridge tectonics appear to provide an efficient

pathway for ascending magma. Thus, the low-angle, thrust fault in Fig. 3 formed during the faulting and folding of upper basaltic layers, caused by reactivation of sub-mare normal faults under a dominant contractional stress field.] Is the oblique/slip fault that connects to the normal faults created by the formation of the WRs or is it created by having to sets of WR like in the transfer zone of the Lambert region? [Sorry for the confusion! The oblique/slip fault connects to the sub-mare basement-involved normal faults due to reactivation of the normal faults under a dominant contractional stress field.]

8. If the formation of WR re-activates older normal faults, does it mean the magma within the normal faults had to be warm/buoyant enough to ascend through the newly formed faults? [Both the reactivation of older normal faults and resulting WR formation, under a dominant contractional stress field, provide conduits for **later** magma transport. When the abovementioned tectonics happened, the magma chamber should have already existed in the underlying upper mantle or crust/mantle interface (Head and Wilson, 2017). The reactivation tectonics opens channels for the magma ascent and hence erupt. It should be noted that reactivation may have occurred in many locations, but not all of them are related to volcanism.] Is there any constraint on how far apart in time a normal fault had to be from the formation of the WR? [The normal faults formed early after the impact due to transient collapses and subsequent tectonic adjustment, but later subsidence of basaltic fillings and underlying accumulation of magma to inflate its overlying lithosphere will cause ring/rim normal faults to dilate, and then at some locations may be reused for magma transport. The reactivation of normal faults during inversion tectonics, during which WRs formed, should have occurred when the Moon's environment dominated by compressional stresses, in a striking contrast with early inflationary expansion stage.]

9. Lines 121-123 mention that mare emplacement caused compression within the Lambert-SE region. How can mare emplacement cause compression? [Mare emplacements cause mass loading of dense basalts in the basin interior, with the thickness decreasing from the basin center to its margin. Then, the tectonic adjustment of mascon and basalt loading produce compressional stress in the interior and extensional stress to its exterior. According to existing geophysical models (Melosh et al., 1978; Solomon and Head, 1979, 1980; Solomon, 1986; Freed et al., 2001), the Lambert-SE region is influenced by the E-W compression, and the direction of the maximum horizontal stress axis is concentric to the basin center.] I understand that a thicker emplacement of mare within the basin center cause a pressure gradient in that direction, but I don't understand why the emplacement result in compression. [Here, we did not mean the mare emplacement only confined to the Lambert area, but after the sequences of early mare emplacement-derived loading for the whole basin.] For compression, two forces with opposite directions need to be collide within the Lambert area. [That's the case for the Lambert area. Please see Fig. S3 for the regional stress analysis and Fig. 2 in Melosh (1978) for the "strike slip faults (SS)" tectonic style.]

REVIEWERS' COMMENTS

Reviewer #2 (Remarks to the Author):

Review: Revision of Evidence for Structural Control of Mare Volcanism in Lunar Compressional Tectonic Settings

By: Feng Zhang, Alberto Pizzi, Trishit Ruj, Goro Komatsu, An Yin, Yang Liu, Yongliao Zou

The revised paper investigates the relation between contractional wrinkle ridges and volcanic landforms in the mare basalts. It is suggested that the paradox of contemporaneous contraction of the mare basalts with continued mare volcanic activity can be explained by localization of wrinkle ridges over basement-involved normal faults. The authors suggest that reactivated buried, deeply rooted, normal faults provided conduits for magma ascent.

The authors have made a concerted effort to address the comments of the reviewers, and the paper is significantly improved, particularly since strike-slip faults are now not the principal focus. However, some weaknesses remain. There is now a reliance on basement-involved ring/rim normal faults to localize mare wrinkle ridges. Not all the patterns of mare wrinkle ridges can be attributed to localization by preexisting basement structures, particularly those in non-mascon mare settings that may not have the basement structures expected in mascon basins. The authors must make this clear in the paper. The authors should also recognize that en echelon arrangements are very common with wrinkle ridges and nearside graben. Such arrangements are how faults link and grow (see Martin and Watters, Icarus, 2022). Volcanism may also be exploiting linkage zones without any strike-slip involvement.

A greater problem is that the authors are arguing from the specific cases examined to the general. Only 30 cases of wrinkle ridges with associated volcanic feature are examined. This is a small sample of the many hundreds of wrinkle ridges in mare basalts. The suggestion that these relatively few examples can be used to make a case for all mare wrinkle ridges forming by interaction of faults and magma is flawed. The cited examples may well be cases of volcano-tectonic interactions, but there is no evidence to support the conclusion that the vast majority of mare wrinkle ridges are localized by basement-involved normal faults. I recommend the authors restrict their conclusions to the class of examples studied and avoid generalizing from the relatively few cases to all mare wrinkle ridges.

Thus, although the revision is improved, it still suffers from some weaknesses. I would support a revised paper limited to, and focused on, the specific cases examined for publication in Nature Communications. I do not recommend the authors propose broad reexamination of models for mare evolution that involve some cadence between thermal expansion and contraction that is not predicted in thermal history models or observed outside of the cases studied.

Other suggestions are given below.

Line 250, The suggestion that mare wrinkle ridges in general are the result of tectonic inversion is not supported by the data. There are many cases, particularly in non-mascon mare settings, where there is no evidence of basement-involved structures that localize ridge formation. This is an over generalization.

Line 337, The conventional wisdom as to why the wrinkle ridge style of deformation is confined solely to the mare is because the mare basalts sequences are a multilayer and deform as a multilayer rather than a mechanically isotropic material like the highlands. This is clearly demonstrated by wrinkle ridge-lobate scarp transitions that occur at mare-highland contacts (see Watters and Johnson, 2010; Williams et al., Icarus, 2019).

Line 379, The statement that "The current models of the volcanic and tectonic evolution of mascon basins (i.e., lunar thermal evolution) must be reviewed and reconsidered with the above-mentioned considerations given that the lunar thermal expansion and cooling contraction are

always treated separately" is not justified. Thermal history models for the Moon do not support periods of late stage thermal expansion. Also, basin-localized extension was inferred to have ceased ~3.6 Ga ago (Lucchitta and Watkins 1978). The stress state of the Moon has had a compressional stress bias due to ongoing interior cooling since then - as predicted by thermal history models (e.g., Pritchard and Stevenson, 2000).

Line 385, "The apparent relationship that all the mare wrinkle ridges should postdate their host mare units may be not always true because wrinkle ridge-formation compressional tectonics can also trigger later eruptions." This conclusion is also difficult to justify. It is not at all clear how contractional tectonics in the mare would trigger later volcanic eruptions.

[REDACTED]

Reviewer #3 (Remarks to the Author):

My review corresponds to the revised version of the article. I originally agreed with the relevance of the work but questioned how strongly the conclusions were supported by the authors. My original main comment was:

This paper proposes a mechanism to explain the relationship of wrinkle ridges (WR) to volcanic features in the Moon. In consequence, this mechanism would also explain how volcanism can exist in compressional settings. If this mechanism is correct, it would significantly impact our current understanding of the thermal history of the Moon because compression on the Moon would not be due exclusively to cooling after mare emplacement, but compression could instead occur during mare emplacement periods. In general, the promise of the paper is important for the lunar volcanism science community, however, some details need to be improved so the paper can support its results and conclusions. I recommend authors work on the following comments and the paper gets a second round of reviews before it can be published.

The revised version of the manuscript contains an improved description of the proposed mechanism of wrinkle formation and relationship to the tectonics of the mare emplacement. Additionally, the revised version puts the results of the paper in context with the timeline of the mascon formation. In general, the authors improve the clarity of the text and provide further references that support their results.

I think the work is relevant and the conclusions are well justified. In my opinion, the work is ready to be published.

REVIEWERS' COMMENTS

Reviewer #2 (Remarks to the Author):

Review: Revision of Evidence for Structural Control of Mare Volcanism in Lunar Compressional Tectonic Settings

By: Feng Zhang, Alberto Pizzi, Trishit Ruj, Goro Komatsu, An Yin, Yang Liu, Yongliao Zou

The revised paper investigates the relation between contractional wrinkle ridges and volcanic landforms in the mare basalts. It is suggested that the paradox of contemporaneous contraction of the mare basalts with continued mare volcanic activity can be explained by localization of wrinkle ridges over basement-involved normal faults. The authors suggest that reactivated buried, deeply rooted, normal faults provided conduits for magma ascent.

The authors have made a concerted effort to address the comments of the reviewers, and the paper is significantly improved, particularly since strike-slip faults are now not the principal focus. However, some weaknesses remain. There is now a reliance on basement-involved ring/rim normal faults to localize mare wrinkle ridges. Not all the patterns of mare wrinkle ridges can be attributed to localization by preexisting basement structures, particularly those in non-mascon mare settings that may not have the basement structures expected in mascon basins. [We agree the reviewer's point.] The authors must make this clear in the paper. [We have made this clear in the main text with changing the beginning title of the paragraph to be like this: Tectonic inversion origin for the thrust-related mare volcanism within mascon basin tectonic domains.] The authors should also recognize that en echelon arrangements are very common with wrinkle ridges and nearside graben. Such arrangements are how faults link and grow (see Martin and Watters, Icarus, 2022). [We agree that the en echelon pattern is the natural way of faults and folds to develop and grow, however it is notable the systematic left stepping arrangements of the (N)NW-striking WRs in the Lambert-SE area, suggest a robust structural consistency with the estimated stress field having the maximum horizontal compressional axis oriented approximately E-W.] Volcanism may also be exploiting linkage zones without any strike-slip involvement. [Thanks, and we agree the reviewer's points. We will consider this carefully during our efforts to explore more and individual cases in future studies.]

A greater problem is that the authors are arguing from the specific cases examined to the general. [We did not mean so, but we have specified this and made it clear in some places (in particular the first section of the 'Discussion' part) that our proposed model currently support the cases in zones of mascon-basin rings/rims with distributed normal faults in the basement crust.] Only 30 cases of wrinkle ridges with associated volcanic feature are examined. This is a small sample of the many hundreds of wrinkle ridges in mare basalts. [Yes, indeed, it is. But, we did not perform global search for all of them. We believe that more cases will be identified in future efforts. In addition, with analogue studies of their

terrestrial counterparts, we believe that more than the 30 cases would have resulted from the same mechanism, but not all of them concurrent with mare volcanism. Further, the identified 30 cases, after all, are not a small number considering that the volcanic eruptive center-hosting mare ridges cover (or cut across) a significantly large area of many mascon basin mares.] The suggestion that these relatively few examples can be used to make a case for all mare wrinkle ridges forming by interaction of faults and magma is flawed. [Thanks for pointing this. We have gone through the whole manuscript, and tried to avoid including all mare wrinkle ridges.] The cited examples may well be cases of volcano-tectonic interactions, but there is no evidence to support the conclusion that the vast majority of mare wrinkle ridges are localized by basement-involved normal faults. I recommend the authors restrict their conclusions to the class of examples studied and avoid generalizing from the relatively few cases to all mare wrinkle ridges. [We have done so, as suggested.]

Thus, although the revision is improved, it still suffers from some weaknesses. I would support a revised paper limited to, and focused on, the specific cases examined for publication in Nature Communications. I do not recommend the authors propose broad reexamination of models for mare evolution that involve some cadence between thermal expansion and contraction that is not predicted in thermal history models or observed outside of the cases studied. [Thanks for the reviewer's constructive comments and useful suggestions. We have put our focus on the specific cases investigated and slightly spread this to a certain subset (limited to the zones of mascon basin rings/rims) of all the mare wrinkle ridges. This is because: (1) more cases will be found if future global searching efforts are performed; (2) more wrinkle ridges with the exception of the 30 investigated cases may have formed resulting from the proposed mechanism (i.e., tectonic inversion), but for most of them, the WR-formation compressional tectonics may have not been accompanied by mare volcanism. We have specified this in the main text: lines 290-294.]

Other suggestions are given below.

Line 250, The suggestion that mare wrinkle ridges in general are the result of tectonic inversion is not supported by the data. There are many cases, particularly in non-mascon mare settings, where there is no evidence of basement-involved structures that localize ridge formation. This is an over generalization.

Thanks for pointing out this. Actually, we did not mean to make the explanation spread to all the mare wrinkle ridges. To make it clear, we have specified this like below: "Tectonic inversion origin for the thrust-related mare volcanism within mascon basin tectonic domains."

Line 337, The conventional wisdom as to why the wrinkle ridge style of deformation is confined solely to the mare is because the mare basalts sequences are a multilayer and deform as a multilayer rather than a mechanically isotropic material like the highlands. This is clearly demonstrated by wrinkle ridge-lobate scarp transitions that occur at mare-

highland contacts (see Watters and Johnson, 2010; Williams et al., Icarus, 2019).

Thanks for the information! This knowledge background has been considered and added in our newly revised manuscript (Lines 371-375).

Line 379, The statement that “The current models of the volcanic and tectonic evolution of mascon basins (i.e., lunar thermal evolution) must be reviewed and reconsidered with the above-mentioned considerations given that the lunar thermal expansion and cooling contraction are always treated separately” is not justified. Thermal history models for the Moon do not support periods of late stage thermal expansion. Also, basin-localized extension was inferred to have ceased ~3.6 Ga ago (Lucchitta and Watkins 1978). The stress state of the Moon has had a compressional stress bias due to ongoing interior cooling since then - as predicted by thermal history models (e.g., Pritchard and Stevenson, 2000).

We have reconsidered and thought this carefully. As the reviewer pointed, the previous studies often considered the ~3.6 Ga as a turning point between basin-localized extension and later compression-dominated cooling stage. However, currently more studies indicate that the major emplacement of mare basalts occurred ~3.0 Ga ago (e.g., Head & Wilson, 1992; Hiesinger et al., 2011). The overlapped periods of cooling contraction and continued mare volcanism (between ~3.0 and 3.6 Ga) may support and be well consistent with our proposed model and conclusions: during this transition period, the occurrence of late-stage mare volcanism (also not all of them) is concurrent with mare wrinkle ridge-formation tectonics, which opened vertical channels for magma ascent and eruption. The abovementioned sentence, as highlighted by the reviewer, has been reworked following the reviewer’s comments and instruction (See lines 375-381).

Line 385, “The apparent relationship that all the mare wrinkle ridges should postdate their host mare units may be not always true because wrinkle ridge-formation compressional tectonics can also trigger later eruptions.” This conclusion is also difficult to justify. It is not at all clear how contractional tectonics in the mare would trigger later volcanic eruptions.

This was not intended to say that contractional tectonics in the mare would trigger volcanic eruptions in a long time later. We wanted to express that the occurrence of mare volcanism can be concurrent with WR formation tectonics, and thus some mare basalts would have been emplaced by eruptions with their conduits opened by WR formation tectonics (reworked with your comments, lines 385-388). Thus, the vent-bearing WRs and their surrounding mares are formed at nearly the same time, rather than being a simple order of the former formed after the latter. In addition, from the comments above, the reviewer also pointed out that: volcanism may also be exploiting linkage zones (*en*-echelon pattern faults) of wrinkle ridge segments. If this is true, it tells us that contraction tectonics in the mare can cause later volcanic eruptions because the magmas can use the mare weaknesses associated with WRs as their conduit for later (with long or short time-gap) ascent and eruption to the surface.

[REDACTED]

Reviewer #3 (Remarks to the Author):

My review corresponds to the revised version of the article. I originally agreed with the relevance of the work but questioned how strongly the conclusions were supported by the authors. My original main comment was:

This paper proposes a mechanism to explain the relationship of wrinkle ridges (WR) to volcanic features in the Moon. In consequence, this mechanism would also explain how volcanism can exist in compressional settings. If this mechanism is correct, it would significantly impact our current understanding of the thermal history of the Moon because compression on the Moon would not be due exclusively to cooling after mare emplacement, but compression could instead occur during mare emplacement periods. In general, the promise of the paper is important for the lunar volcanism science community, however, some details need to be improved so the paper can support its results and conclusions. I recommend authors work on the following comments and the paper gets a second round of reviews before it can be published.

Thanks for the originally helpful comments during the first circle of review process, which have helped us to achieve a significant improvement of the original manuscript.

The revised version of the manuscript contains an improved description of the proposed mechanism of wrinkle formation and relationship to the tectonics of the mare emplacement. Additionally, the revised version puts the results of the paper in context with the timeline of the mascon formation. In general, the authors improve the clarity of the text and provide further references that support their results.

Thanks for the very positive comments on our last revised manuscript.

I think the work is relevant and the conclusions are well justified. In my opinion, the work is ready to be published.

Thanks again for the very positive comments and your recommendation: **“the work is ready to be published”**.